# Light-modulated neural control of sphincter regulation in the evolution of through-gut

Junko Yaguchi [1], Kazumi Sakai[2], Atsushi Horiuchi[2], Takashi Yamamoto [3], Takahiro Yamashita [2] & Shunsuke Yaguchi [1,4] ✉

The development of a continuous digestive tract, or through-gut, represents a key milestone in bilaterian evolution. However, the regulatory mechanisms in ancient bilaterians (urbilaterians) are not well understood. Our study, using larval sea urchins as a model, reveals a sophisticated system that prevents the simultaneous opening of the pylorus and anus, entry and exit points of the gut. This regulation is influenced by external light, with blue light affecting the pylorus via serotonergic neurons and both blue and longer wavelengths controlling the anus through cholinergic and dopaminergic neurons. These findings provide new insights into the neural orchestration of sphincter control in a simplified through-gut, which includes the esophagus, stomach, and intestine. Here, we propose that the emergence of the earliest urbilaterian through-gut was accompanied by the evolution of neural systems regulating sphincters in response to light, shedding light on the functional regulation of primordial digestive systems.

Efficient nutrient assimilation is crucial for the survival of organisms, suggesting that digestive systems have undergone evolutionary refinement tailored to specific life histories. One of the most significant evolutionary developments in this context is the emergence of the through-gut in ancient bilaterians, or urbilaterians. This innovation likely provided a major advantage by significantly enhancing digestive efficiency and nutrient absorption, a stark contrast to the gastrovascular cavity of the common ancestors of cnidarians and bilaterians, which featured a single orifice serving both as mouth and anus, as observed in extant cnidarians[1–3]. Non-bilaterian metazoans generally lack a through-gut, though ctenophores exhibit oral and anal structures, whose homology with the bilaterian tripartite gut system is debated[4].

This raises questions about the evolution of functional through-guts. Xenacoelomorpha, a group without a through-gut, plays a pivotal role in this discussion. Phylogenomic analyses suggest Xenacoelomorpha may be sister to Nephrozoa, implying their simple body plans might reflect either ancestral traits or secondary simplification[5–8]. Clarifying their phylogenetic position is essential for understanding the evolution of the through-gut and other bilaterian features in future.

Despite the clear evolutionary significance of the through-gut, the mechanisms driving its development in urbilaterians remain enigmatic[1,9]. Current research is focused on unraveling these mechanisms, with model organisms like sea urchin larvae providing valuable insights. These larvae, which belong to deuterostomes within bilaterians, exhibit a tripartite through-gut architecture analogous to that of vertebrates (Fig. 1a). Their gene expression profile along the digestive tract mirrors that of vertebrates, making them an ideal model for studying gut morphogenesis and functionality[10]. The transparent bodies of sea urchin larvae, combined with the absence of most muscular structures aside from those in the digestive tract, facilitate in vivo observation and precise analysis of internal processes[11,12]. This simplicity makes sea urchins promising for understanding the regulatory mechanisms of the through-gut, likely present in urbilaterians or at least in the common ancestor of deuterostomes.

In this context, understanding the nervous system's role in regulating the digestive tract is equally crucial. Previous studies have provided insights into the nervous system of echinoderm larvae, particularly focusing on the well-studied sea urchin larvae. In the

[1]Shimoda Marine Research Center, University of Tsukuba, 5-10-1 Shimoda, Shizuoka 415-0025, Japan. [2]Department of Biophysics, Graduate School of Science, Kyoto University, Kitashirakawa-Oiwake, Sakyo-ku, Kyoto, Kyoto 606-8502, Japan. [3]Graduate School of Integrated Sciences for Life, Hiroshima University, Higashi-Hiroshima, Hiroshima 739-8526, Japan. [4]Japan Science and Technology Agency, PRESTO, 7 Gobancho, Chiyoda-ku 102-0076 Tokyo, Japan. ✉ e-mail: yag@shimoda.tsukuba.ac.jp

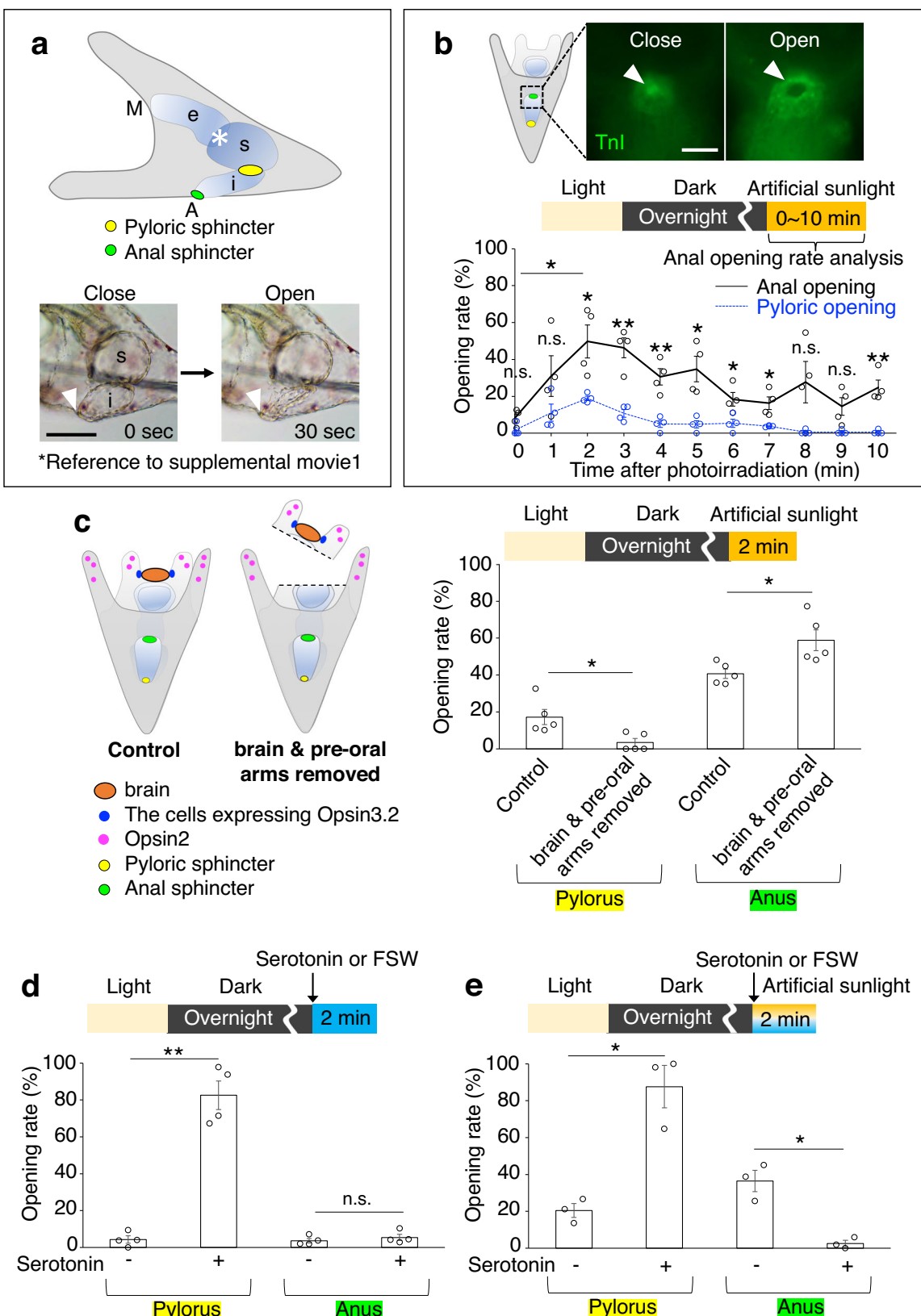

anterior part of the larvae, defined as the brain based on its gene expression profile and its role in integrating external stimuli and transducing these signals into behavioral responses[13,14], serotonergic and cholinergic neurons are present[15,16]. Many other neurons are located in the ciliary band region and are either cholinergic or dopaminergic[17–20].

While there are not many experimental studies on their functions, some reports suggest involvement in the control of the digestive tract and swimming[14,17,21]. Each neuron extends long axons but is implied to form no synaptic structures[22]. In vertebrates, the enteric nervous system[23–25], digestive tract secretory cells[26], and the vagus nerve from the brain control the digestive process[27,28]. Enterochromaffin cells in

**Fig. 1 | Light-induced anus opening in sea urchin larvae. a** Lateral view of a sea urchin larva with the digestive tract illuminated, showing key anatomical features: the cardiac sphincter (*, asterisk), pyloric sphincter (yellow), and anal sphincter (green). The inset on the bottom left depicts the anus in a closed state (arrowhead), while the bottom right shows the anus open and the intestine (i) contracted, 30 seconds post-photoirradiation. Annotations: M, mouth; e, esophagus; s, stomach; i, intestine. **b** Anus opening rates post-photoirradiation, assessed using anti-Troponin I (TnI) antibody staining (refer to Methods). The graph presents basic photoirradiation conditions alongside opening rates, with pyloric (blue) and anal (black) openings (N = 4 batches [each consisting of a different male and female pair], n [0 min] = 27, 32, 36, 46 larvae, n [1 min] = 18, 23, 43, 33 larvae, n [2 min] = 18, 37, 30, 45 larvae, n [3 min] = 14, 24, 42, 49 larvae, n [4 min] = 15, 11, 34, 39 larvae, n [5 min] = 21, 20, 40, 42 larvae, n [6 min] = 21, 25, 45, 36 larvae, n [7 min] = 24, 26, 27, 30 larvae, n [8 min] = 16, 27, 44, 40 larvae, n [9 min] = 20, 24, 52, 45 larvae, n [10 min] = 19, 25, 41, 44 larvae; pylorus: [0 min] mean 1.6% ± 1.6% SEM, [1 min] mean 11.1% ± 4.7% SEM, [2 min] mean 18.9% ± 1.2% SEM, [3 min] mean 10.8% ± 2.1% SEM, [4 min] mean 5% ± 1.9% SEM, [5 min] mean 4.8% ± 1.9% SEM, [6 min] mean 5.4% ± 2.3% SEM, [7 min] mean 3.8% ± 0.2% SEM, [8 min] mean 0.6% ± 0.6% SEM, [9 min] mean 0.6% ± 0.6% SEM, [10 min] mean 0.6% ± 0.6% SEM; anus: [0 min] mean 8.2% ± 2.2% SEM, [1 min] mean 31% ± 11% SEM, [2 min] mean 49.7% ± 8.9% SEM, [3 min] mean 46.3% ± 5.4% SEM, [4 min] mean 30.6% ± 4.4% SEM, [5 min] mean 34.8% ± 6.9% SEM, [6 min] mean 18.4% ± 3.7% SEM, [7 min] mean 16.3% ± 3.5% SEM, [8 min] mean 27.7% ± 11.2% SEM, [9 min] mean 14.5% ± 4.8% SEM, [10 min] mean 24.8% ± 4.1% SEM). **c.** Schematic of the experimental setup brain and pre-oral arm ablation with a needle. Color coding: orange, brain; blue, Opsin3.2-expressing cells; magenta, Opsin2-expressing cells; yellow, pylorus; green, anus. The graph illustrates the opening rates of the pylorus and anus 2 minutes after photoirradiation (N = 5 batches, n [control] = 95, 38, 54, 49, 37 larvae, n [brain & pre-oral arms removed] = 54, 22, 27, 28, 25 larvae; pylorus: [control] mean 17.2% ± 4.1% SEM, [brain & pre-oral arms removed] mean 3.5% ± 2.1% SEM; anus: [control] mean 40.7% ± 2.5% SEM, [brain & pre-oral arms removed] mean 58.8% ± 5.7% SEM). **d, e** Graphs depicting the opening rates of the pylorus and anus under various conditions: **d** serotonin treatment (10 μM) without photoirradiation (N = 4 batches, n [serotonin -] = 53, 42, 49, 53 larvae, n [serotonin +] = 35, 48, 45, 58 larvae; pylorus: [serotonin -] mean 4.3% ± 2% SEM, [serotonin +] mean 82.5% ± 7.7% SEM; anus: [serotonin -] mean 3.7% ± 1.2% SEM, [serotonin +] mean 5.5% ± 1.7% SEM); **e** serotonin treatment (10 μM) with photoirradiation (N = 3 batches, n [serotonin -] = 147, 71, 94 larvae, n [serotonin +] = 53, 31, 34 larvae; pylorus: [serotonin -] mean 20.4% ± 3.8% SEM, [serotonin +] mean 87.6% ± 11.5% SEM; anus: [serotonin -] mean 36.5% ± 5.8% SEM, [serotonin +] mean 2.6% ± 1.7% SEM). Statistical significance denoted as *p < 0.05, **p < 0.01; n.s. = not significant, Welch's *t* test (two-sided). Error bars shown in all graphs indicate SEM. Scale bar in a = 50 μm and in **b** = 20 μm. Source data are provided as a Source Data file.

the gut contain large amounts of serotonin, crucial for maintaining digestive function[29], while nitric oxide neurons contribute to proper pyloric formation during gut development[30,31]. Intriguingly, in sea urchin larvae, the enteric nervous system also aids in pyloric formation, despite the absence of serotonin in their enteric system[32]. Instead, brain serotonin controls pyloric opening in response to light[14]. This suggests that in echinoderms, including sea urchins, various nerves secrete neurotransmitters hormonally into the coelomic cavity, covering a broad range[22].

The evolution of the through-gut represents a pivotal development in the history of metazoans, with significant implications for digestive efficiency and nutrient absorption. While much has been learned, the precise mechanisms underlying this evolutionary innovation continue to be a rich area for research, promising to yield deeper insights into the complex interplay between anatomy, genetics, and physiology in early bilaterians. In our previous work, we discovered and reported that light exposure influences the opening of the pyloric sphincter in sea urchin larvae, thereby delineating the neural pathway from photoreception to sphincter modulation[14].

Herein, we present findings on the regulatory systems governing both the entrance and exit of sea urchin larval intestine, offering insights into the integrated control mechanisms of the gastrointestinal cavity. Given the pylorus's role as the intestinal gateway, our current study shifts focus to the anal sphincter, the intestinal exit. Especially, observations conducted under a dissecting microscope revealed a marked increase in the number of excrements per embryo 10 minutes after being transferred to light condition from dark, indicating that sea urchin larvae initiate defecation in response to light stimuli (Supplementary Fig. 1a).

## Results
### Light exposure induces anus opening
Here, we report that the anus of sea urchin larvae starts to open within 1 to 2 minutes after being irradiated with light. Initially, larvae were subjected to photoirradiation using artificial sunlight, fixed at predetermined intervals, and subsequently stained with the muscle-specific antibody anti-Troponin-I[11] (Fig. 1b). This procedure was employed to facilitate precise observation of the anal opening, analogous to our previously reported findings on the pyloric opening[14].

Directly determining the opening and closing of the anus in live larvae is challenging; reliance on intestinal contraction provides only an indirect assessment (Fig. 1a, white arrowheads). Furthermore, we employed a food-free protocol in this experiment to minimize the stimulatory effects of food on gastrointestinal activity. This approach was aimed at isolating the intrinsic/background mechanisms of digestive function modulated by light input, as previously discussed[14].

The timing of anal opening varied among individuals, and the frequency of anal opening was significantly higher compared to that of the pylorus during 2 to 7 min of the 10-minute observation period (Fig. 1b). A peak occurrence was observed at two minutes post-photoirradiation, with nearly all larvae exhibiting anal opening events several times within a 10-minute window under artificial sunlight exposure (Fig. 1a, b, Supplementary movie 1). Therefore, as previously described[14], we set a constant light/dark cycle (10 min light, 16 h dark, and 2 min with or without photoirradiation [photon flux density, 1000 μmol m-2 s-1]). We fixed the larvae 2 min after photoirradiation and checked the opening and closing of the pyloric and anal sphincters using immunohistochemistry with anti-TroponinI, which specifically detects these sphincters. The ratio of pyloric and anal opening refers to the percentage of the population of larvae with open sphincters, and was quantified in fixed and immunostained larvae. The anal opening is not induced by red light, allowing us to use red light when applying reagents in subsequent experiments (Supplementary Fig. 1b).

To elucidate the pathway governing anal opening, we first excised the brain region and assessed the rate of anal opening under photoirradiation. This approach was informed by our previous studies, which identified that Opsin3.2 and serotonergic neurons in this region play a pivotal role in regulating pyloric opening[14]. Larvae devoid of brain tissue exhibited a marked reduction in pyloric opening, as previously reported, but displayed a slight increase, rather than a decrease, in anal opening (Fig. 1c). Notably, at this developmental stage, the predominant neuronal type in the brain is serotonergic[33]; however, serotonin was found to suppress, rather than induce, anal opening, in contrast to its effect on the pylorus[14] (Fig. 1c–e, Supplementary Fig. 2a, b). Furthermore, selective attenuation of brain-related Opsin3.2[34,35] had no appreciable effect on this process (Supplementary Fig. 2c), suggesting distinct mechanisms underlie the regulation of anal and pyloric openings.

Subsequently, our focus shifted to another larval Opsin, Opsin2, expressed in the larval arms[17]. We inhibited its function through specific morpholino-mediated knockdown (Fig. 2a, b Supplementary Fig. 3a–c). Consequently, anal opening was incomplete even under intense photoirradiation, while the pyloric response remained unaffected (Fig. 2b). This observation aligns with previous reports indicating that Opsin2 plays a role in inhibiting cholinergic neurons during photoirradiation[17]. Inhibition of cholinergic neurons using Atropine[17], a

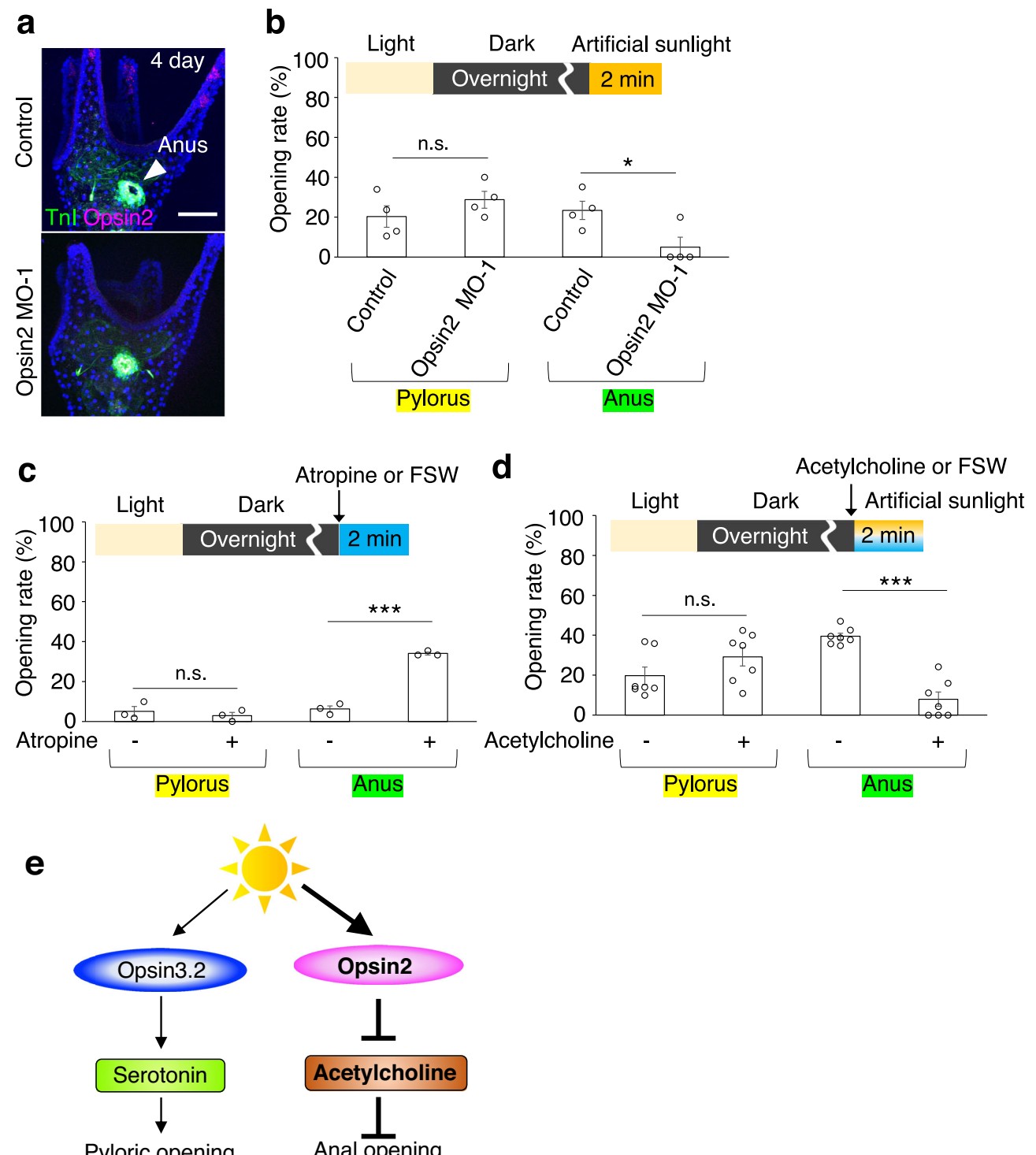

**Fig. 2 | Light-induced anus opening is mediated by Opsin2. a** Opsin2 cells are missing in Opsin2 morphants. Anus opening or closing is identified with anti-TroponinI (TnI) antibody (arrowhead). **b**–**d** Graphs depicting the opening rates of the pylorus and anus under various conditions: **b** in control versus Opsin2 morphants post-photoirradiation ($N = 4$ batches [each consisting of a different male and female pair], $n$ [control] = 38, 77, 53, 19 larvae, $n$ [Opsin2 MO-1] = 16, 10, 10, 20 larvae; pylorus: [control] mean 20.3% ± 5.4% SEM, [Opsin2 MO-1] mean 28.8% ± 4.3% SEM; anus: [control] mean 23.5% ± 4.5% SEM, [Opsin2 MO-1] mean 5% ± 5% SEM); **c** with or without atropine treatment (100 μM) ($N = 3$ batches, $n$ [atropine -] = 57, 118, 61 larvae, $n$ [atropine +] = 18, 42, 31 larvae; pylorus: [atropine -] mean 5% ± 2.5% SEM, [atropine +] mean 2.9% ± 1.6% SEM; anus: [atropine -] mean 6.2% ± 1.6% SEM,

[atropine +] mean 34.1% ± 0.7% SEM); **d** with or without acetylcholine treatment (1 μM) with photoirradiation ($N = 7$ batches, $n$ [acetylcholine -] = 38, 104, 70, 115, 114, 92, 147 larvae, $n$ [acetylcholine +] = 46, 23, 40, 36, 33, 25, 62 larvae; pylorus: [acetylcholine -] mean 19.7% ± 4.3% SEM, [acetylcholine +] mean 29.2% ± 4.6% SEM; anus: [acetylcholine -] mean 39.5% ± 1.6% SEM, [acetylcholine +] mean 8% ± 3.6% SEM). **e** Illustration of the signaling pathways from light perception to pyloric and anal opening, mediated by Opsin proteins and neurotransmitters. Statistical significance denoted as *$p < 0.05$, ***$p < 0.001$; n.s. = not significant, Welch's $t$ test (two-sided). Error bars shown in all graphs indicate SEM. Scale bar in **a** = 20 μm. Source data are provided as a Source Data file.

muscarinic acetylcholine (ACh) receptor antagonist, resulted in the opening of the anus independent of light stimulation (Fig. 2c). Conversely, an excess of ACh hindered anal opening even in the presence of artificial sunlight (Fig. 2d), suggesting that cholinergic neurons inherently obstruct anal opening (Supplementary Fig. 3d–h). Therefore, the suppression of cholinergic neuronal activity via photoreception by Opsin2 appears crucial for facilitating anal opening (Fig. 2e). In summary, light stimulation prompts both pyloric and anal opening in sea urchin larvae, with the Opsin3.2/serotonin and Opsin2/ACh pathways respectively regulating these processes (Fig. 2e).

**Table 1 | The pylorus and the anus do not open simultaneously**

|  |  | Opening rates | The number of opening (N = 38 batches, n = 2778 larvae) |
|---|---|---|---|
| Pyloric opening |  | 19.11% | 531 |
| Anal opening |  | 34,85% | 968 |
| Both of pyloric and anal opening | Experimental | 0.32% | 9 |
|  | Calculated | 6.66% | Approximately 185 |

## Light opens both anus and pylorus, but not simultaneously

During our investigation into the light-responsive behavior of the anus and pylorus in sea urchin larvae, we observed a notable phenomenon: both structures open in response to photoirradiation, but they rarely do simultaneously (Table 1). Approximately 19% of larvae exhibited an open pylorus, while around 35% demonstrated an open anus two minutes following intense photoirradiation. If these openings occurred randomly, it is calculated that about 6.66% of larvae exhibit simultaneous opening of both sphincters. However, our observations deviated significantly from this expectation, with only 0.32% of larvae displaying concurrent opening. In fact, among over 2700 observed larvae, only 9 individuals exhibited simultaneous opening of both sphincters. This strongly implies an interconnected regulatory mechanism between the light-responsive pathways of these two structures. Given that serotonin inhibits anal opening even under photoirradiated conditions (Fig. 1e), it appears that active serotonergic neurons may interfere with the Opsin2-anus pathway. However, this alone does not fully explain the observed phenomena, suggesting the involvement of additional neuronal pathways in this regulation.

In our comprehensive analysis of neurotransmitter synthase gene expression patterns, we identified the expression of choline acetyltransferase (ChAT) at the anus (Fig. 3a–f) and tyrosine hydroxylase

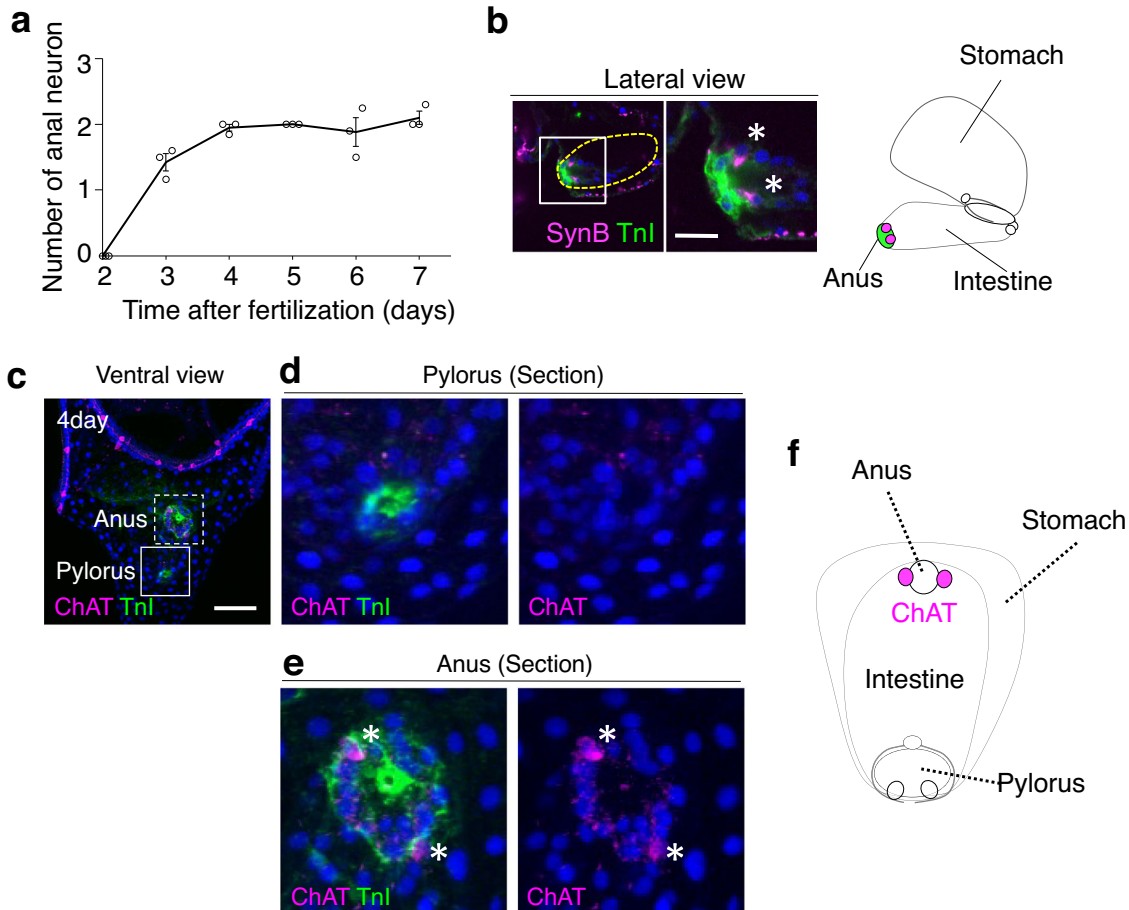

**Fig. 3 | ChAT-positive anal neurons in sea urchin larvae. a** The graph illustrates the number of anal neurons over time, showing the presence of approximately two neurons at the anus starting three days post-fertilization. (N = 3 batches [each consisting of a different male and female pair], n [2-day] = 4, 4, 3 larvae, n [3-day] = 6, 5, 6 larvae, n [4-day] = 8, 13, 3 larvae, n [5-day] = 4, 7, 3 larvae, n [6-day] = 8, 10, 10 larvae, n [7-day] = 7, 10, 8 larvae; [2 day] mean 0% ± 0% SEM, [3 day] mean 1.4% ± 0.1% SEM, [4 day] mean 1.9% ± 0.1% SEM, [5 day] mean 2% ± 0% SEM, [6 day] mean 1.9% ± 0.2% SEM, [7 day] mean 2.1% ± 0.1% SEM). **b** Immunostaining images of anal and intestine (yellow dotted-line) region with anti-SynB antibody (neurons) and anti-TroponinI (TnI) antibody (muscles). A rectangle indicated the magnified region shown in the right. The asterisks in the magnified view highlight the neurons. The schematic diagram presents a lateral view of the stomach and intestine. Magenta cells and green areas indicate neurons and sphincters, respectively, at anus. **c–f** Distribution of cholinergic neurons within the sea urchin larval intestine. **d** ChAT-neurons are not present around the pylorus. **e** Asterisks highlight neurons surrounding the anus. **f** Magenta cells in the schematic image depict ChAT-positive anal neurons. Scale bar in **b** =10 μm; **c** =20 μm. Source data are provided as a Source Data file.

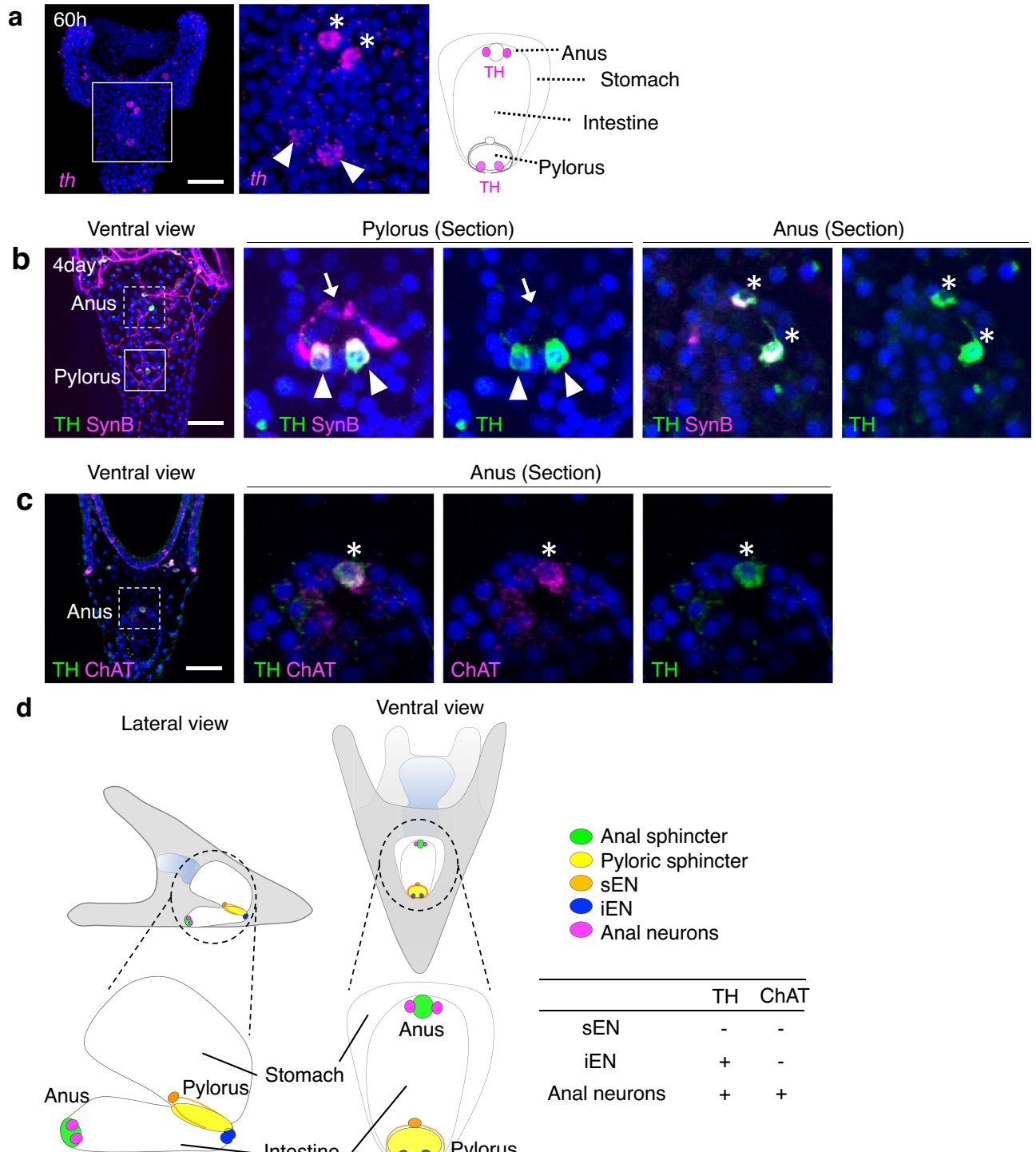

**Fig. 4 | TH-positive pyloric and anal neurons in sea urchin larvae. a** The expression pattern of tyrosine hydroxylase (TH) mRNA in 60-hour larvae. TH is a rate-limiting enzyme for dopamine biosynthesis, with dopamine neurons present at the ciliary band, pylorus (arrowheads), and anus (asterisks). Magenta cells in the schematic image represent dopamine neurons at the pylorus and anus. **b** Arrows indicate stomach-side enteric neurons (sEN); arrowheads point to intestine-side enteric neurons (iEN); asterisks highlight neurons surrounding the anus. **c** Anal neurons (asterisks) co-express ChAT and TH. **d** Schematic representation of the sea urchin larval stomach and intestine, emphasizing neurons near the pylorus (yellow) and anus (green). Neuron types are color-coded: sEN (orange) as TH-/ChAT-, iEN (blue) as TH + /ChAT-, and perianal neurons (magenta) as TH + /ChAT + . Scale bar in **a**, **b**, and **c** =20 μm. Source data for micrograph are provided as a Source Data file.

(TH), a dopamine synthase, at both the anus and pylorus (Fig. 4a–c). The cholinergic and dopaminergic neural patterns in larvae are detailed in Fig. 4d and Supplementary Fig. 4. Excessive dopamine markedly inhibited anal opening but had no significant effect on the pylorus (Fig. 5a, Supplementary Fig. 5a, b). Conversely, the specific inhibition of dopamine receptors facilitated anal opening even in the absence of light (Fig. 5b, Supplementary Fig. 5c–e). These findings strongly indicate that dopamine regularly functions to suppress anal opening, with light stimuli attenuating this effect. Intriguingly, under conditions where dopamine inhibitors (Amisulpride or Eticlopride) are applied, serotonin, a strong inducer of pyloric opening, fails to elicit this response. This suggests that dopamine plays a necessary role in

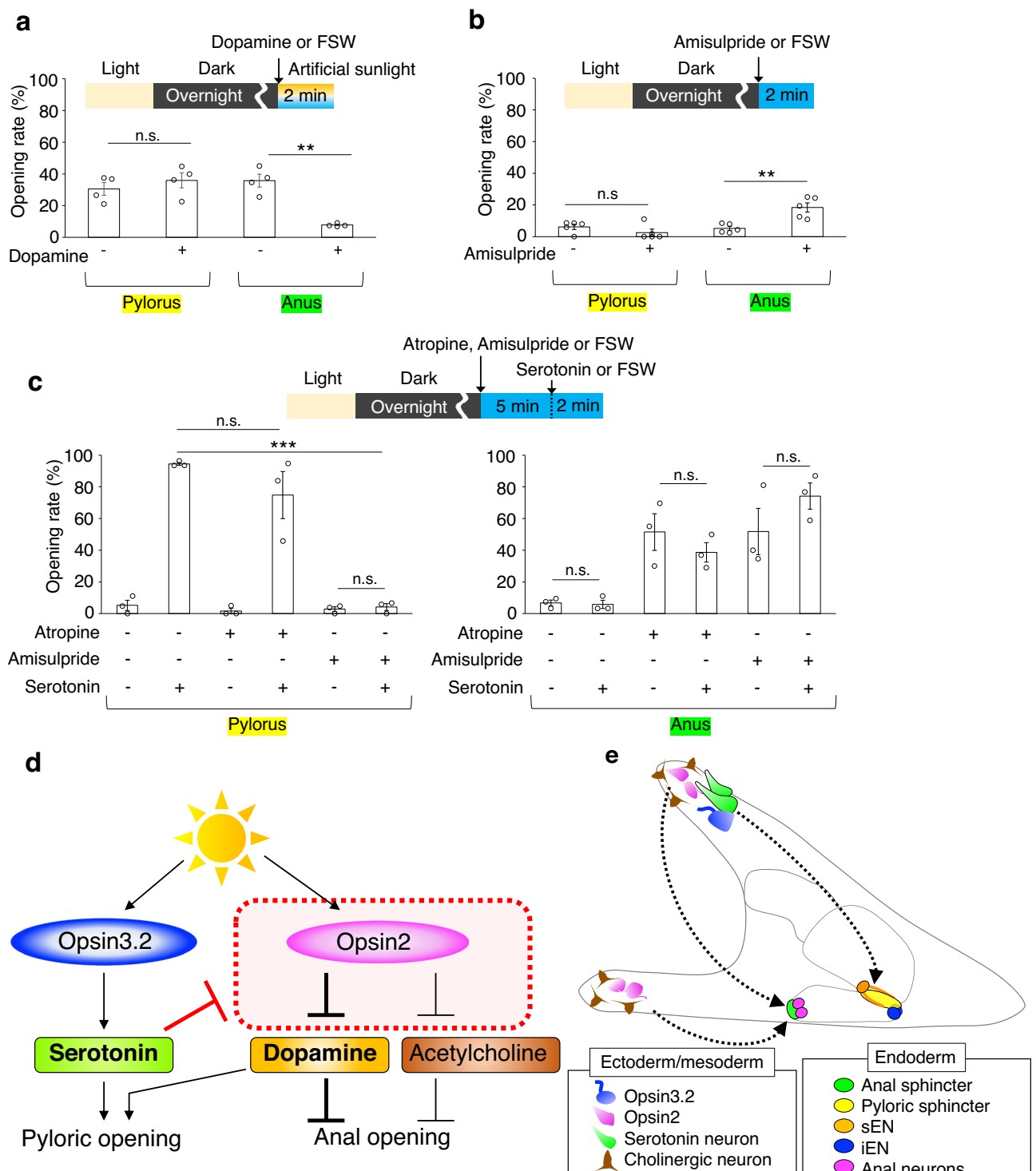

facilitating pyloric opening (Fig. 5c, d, Supplementary Fig. 6a, b), whereas Atropine, ACh receptor antagonist, appears to have no effect on the pylorus. Furthermore, when the activity of either dopamine or ACh, both known inhibitors of anal opening, is disrupted, serotonin loses its ability to suppress anal opening (Fig. 5c). This implies that serotonergic neurons act upstream of dopamine/ACh neurons in this pathway (Fig. 5d), although we cannot completely rule out the possibility that the inhibitors used may block other neural activities in sea urchins.

These findings lead to the hypothesis that rather than the pyloric and anal opening systems directly inhibiting each other in response to the activity of their counterpart, it is the light-responsive pathways that

mutually suppress each other. This mutual suppression establishes a mechanism whereby simultaneous opening of both structures is prevented in sea urchin larvae (Fig. 5d). Such a mechanism is arguably more plausible than one requiring a form of central nervous system-like integration to coordinate muscle movement based on the open/close status of the counterpart. Instead, a system where a single input, such as light, triggers two mutually suppressive pathways is simpler and could have been a feature of the common ancestor of deuterostomes/urbilaterians.

As illustrated in Fig. 5e, we hypothesize the involvement of signaling pathways that activate or suppress neurons surrounding the pyloric or anal sphincters. At this developmental stage, sea urchin larvae

**Fig. 5 | Interactions between pyloric and anal opening pathways in sea urchin larvae. a** Opening rates of the pylorus and anus in larvae treated with or without dopamine (0.1 μM) with photoirradiation ($N = 4$ batches [each consisting of a different male and female pair], $n$ [dopamine -] = 114, 72, 71, 94 larvae, $n$ [dopamine +] = 40, 29, 33, 53 larvae; pylorus: [dopamine -] mean 30.5% ± 4% SEM, [dopamine +] mean 36% ± 4.8% SEM; anus: [dopamine -] mean 35.8% ± 4% SEM, [dopamine +] mean 7.8% ± 0.5% SEM). **b** Opening rates of the pylorus and anus in larvae treated with or without the dopamine inhibitor amisulpride (50 μM) ($N = 5$ batches, $n$ [amisulpride -] = 38, 67, 40, 47, 46 larvae, $n$ [amisulpride +] = 20, 55, 48, 41 36 larvae; pylorus: [amisulpride -] mean 6.2% ± 1.6% SEM, [amisulpride +] mean 2.6% ± 2.2% SEM; anus: [amisulpride -] mean 5.2% ± 1.2% SEM, [amisulpride +] mean 18.4% ± 2.9% SEM). **c** Opening rates of the pylorus and anus in larvae exposed to a combination of inhibitors (100 μM atropine or 50 μM amisulpride) and serotonin (10 μM; applied 5 min after inhibitor treatment), illustrating the modulatory effects on opening behavior ($N = 3$ batches, $n$ [atropine -, amisulpride -, serotonin -] = 27, 21, 30 larvae, $n$ [atropine -, amisulpride -, serotonin +] = 27, 31, 31 larvae, $n$ [atropine +, amisulpride -, serotonin -] = 20, 20, 23 larvae, $n$ [atropine +, amisulpride -, serotonin +] = 19, 31, 24 larvae, $n$ [atropine -, amisulpride +, serotonin -] = 21, 10, 26 larvae, $n$ [atropine -, amisulpride +, serotonin +] = 23, 30, 17 larvae; pylorus: [atropine -, amisulpride -, serotonin -] mean 5.3% ± 3.2% SEM, [atropine -, amisulpride -, serotonin +] mean 94.5% ± 0.9% SEM, [atropine +, amisulpride -, serotonin -] mean 1.7% ± 1.7% SEM, [atropine +, amisulpride -, serotonin +] mean 74.8% ± 14.8% SEM, [atropine -, amisulpride +, serotonin -] mean 2.9% ± 1.5% SEM, [atropine -, amisulpride +, serotonin +] mean 2.9% ± 1.5% SEM, anus: [atropine -, amisulpride -, serotonin -] mean 6.8% ± 1.8% SEM, [atropine -, amisulpride -, serotonin +] mean 5.9% ± 2.6% SEM, [atropine +, amisulpride -, serotonin -] mean 51.5% ± 11.6% SEM, [atropine +, amisulpride -, serotonin +] mean 38.6% ± 6.1% SEM, [atropine -, amisulpride +, serotonin -] mean 51.9% ± 14.6% SEM, [atropine -, amisulpride +, serotonin +] mean 74.1% ± 8.2% SEM). **d** Diagram illustrating the signaling pathways from light perception to pyloric and anal opening, mediated by Opsin proteins and neurotransmitters. The interplay between serotonin and dopamine suggests cross-talk between the pathways. **e** Schematic representation of the putative signaling pathway from photoreceptor cells to the pylorus and anus. Although the exact pathway from Opsin2 to the anus has not yet been identified, it has been reported that the signal from Opsin3.2 photoreception is mediated by serotonergic and nitric oxide (NO) neurons[14]. Opsin2-expressing cells appear to directly connect to cholinergic neurons in the arm region and may also have the potential to secrete unidentified diffusible molecules that mediate the photoreception signal to anal neurons. Statistical significance is indicated as $**p < 0.01$, $***p < 0.001$; n.s. = not significant, Welch's $t$ test (two-sided). Error bars shown in all graphs indicate SEM. Source data are provided as a Source Data file.

possess a few Opsin3.2 photoreceptor cells bilaterally adjacent to the brain region[14]. These cells connect to approximately 5–10 serotonergic neurons[33,36] located in the brain. Our previous study indicated that photoactivation of Opsin3.2 likely directly stimulates these serotonergic neurons, leading to the diffusion of secreted serotonin within the blastocoel, ultimately reaching receptors on the digestive tract. Subsequently, a single nitric oxide-producing neuron (sEN) located at the pyloric sphincter releases nitric oxide, resulting in sphincter muscle relaxation[14,32]. Additionally, two dopamine neurons are situated adjacent to the sphincter[32]; although the precise mechanism remains unclear, these neurons likely stimulate the pyloric sphincter to open. Conversely, within the anal sphincter region, there are one or two anal neurons[22] (Fig. 3a, Supplementary Fig. 4). These neurons co-express ChAT and TH, suggesting a role in the constant suppression of anal opening. The Opsin2 cells, presumed to send inhibitory signals to the anal neurons, located in the arms rather than near the anus, leaving the mediation of these signals unclear. The potential direct connection between Opsin2 cells and cholinergic neurons in the ciliary band may be key to understanding this signaling pathway[17]. Future analyses will be essential to elucidate the detailed molecular signaling involved in the process from photoirradiation to anal opening.

### Light wavelength plays a critical role in determining the selection of alternative pathways for gut regulation

The subsequent question addresses how sea urchin larvae selectively open either the pylorus or the anus in response to a singular stimulus, light. We proposed that the pylorus and anus respond to different effective wavelengths. To test this hypothesis, we employed four types of LED light (460 nm, 520 nm, 555 nm, and 630 nm) and monitored the opening rates of each sphincter two minutes post-photoirradiation. The results revealed a pronounced wavelength dependency for the pylorus with blue light, while the anus responded more broadly to longer wavelengths (Fig. 6a). Complementary in vitro analyses using human cells corroborated these findings; the specific absorbance of Opsin3.2, implicated in pyloric opening, is in the blue region, and the long-wavelength shift of the spectrum after photoreception leads to the suppression of the activity by longer wavelength light irradiation (Fig. 6b, Supplementary Fig. 7a). Conversely, blue light irradiation on Opsin2 induces the spectral shift from the blue region to the violet region (Fig. 6c, see the Figure legend), which is associated with anal opening, and this short-wavelength shift leads to the activation by not only blue light but also longer wavelength light, being supported by blue and green light dependent-elevation of the Ca²⁺ level in Opsin2-expressed cells (Supplementary Figs. 7b, 8).

## Discussion

Based on the data presented, we conclude that the intestinal entrance, the pylorus, opens in response to blue light via the Opsin3.2 and serotonin pathways. Conversely, the intestinal exit, the anus, opens in response to blue/green/yellow light through the inhibitory Opsin2 pathway targeting dopamine, and ACh (Fig. 6d). Serotonin acts to inhibit the pathway linking Opsin2 photoreception and the suppression of dopamine/ACh. This inhibition is crucial for scenarios where the pylorus is open while the anus remains closed. Conversely, under longer wavelength light, the anus opens due to the inhibition of dopamine/ACh, while the pylorus closes as the essential factor, dopamine, is downregulated.

Considering the differing characteristics of opsin proteins, Opsin3.2 and Opsin2 exhibit a long- and short-wavelength shift of the absorption spectrum after photoreception, respectively. The opposite direction of the spectral shift can change the activity of the opsins, both of which absorb blue light, under longer wavelength light and may offer alternative behavioral responses in larvae when exposed to a similar spectral input. This divergence in wavelength sensitivity could potentially influence larval behavior under varying light conditions, enabling a more adaptable response to environmental stimuli.

These mechanisms might align with the diel vertical migration (DVM)[37–42] observed in echinoderm larvae and other zooplankton, where they ascend to the ocean surface at night to feed on phytoplankton. In darkness, the sphincters of the pylorus and anus seldom open, effectively allowing for maximal accumulation of ingested food in the stomach. At dawn, as the larvae descend away from the phytoplankton-rich surface[17], they shift focus from feeding to digestion and nutrient absorption. If the intestine is full, new food cannot enter. Thus, the anus opens first in response to longer wavelengths, which initially illuminate the ocean surface with sufficient photons, facilitating the expulsion of intestinal contents. Although the exact mechanisms governing the closure of the anus and pylorus remain unclear, our findings suggest that the pylorus opens in response to blue light after the anus has closed (Fig. 6d), aligning with the larvae's natural behavioral patterns in their oceanic habitat[12,17].

Intriguingly, the photoreceptors involved in the opening of the anus in sea urchin larvae are consistent with Opsin2, which halts ciliary movement[17]. Typically, sea urchin larvae reside near the ocean's surface, where they coordinate ciliary beating through cholinergic neurons to perform anti-gravity movements, although the detailed mechanisms remain unclear. Opsin2, upon exposure to light, suppresses cholinergic neuronal activity, causing the larvae to sink. This phenomenon has been suggested as a component of the DVM

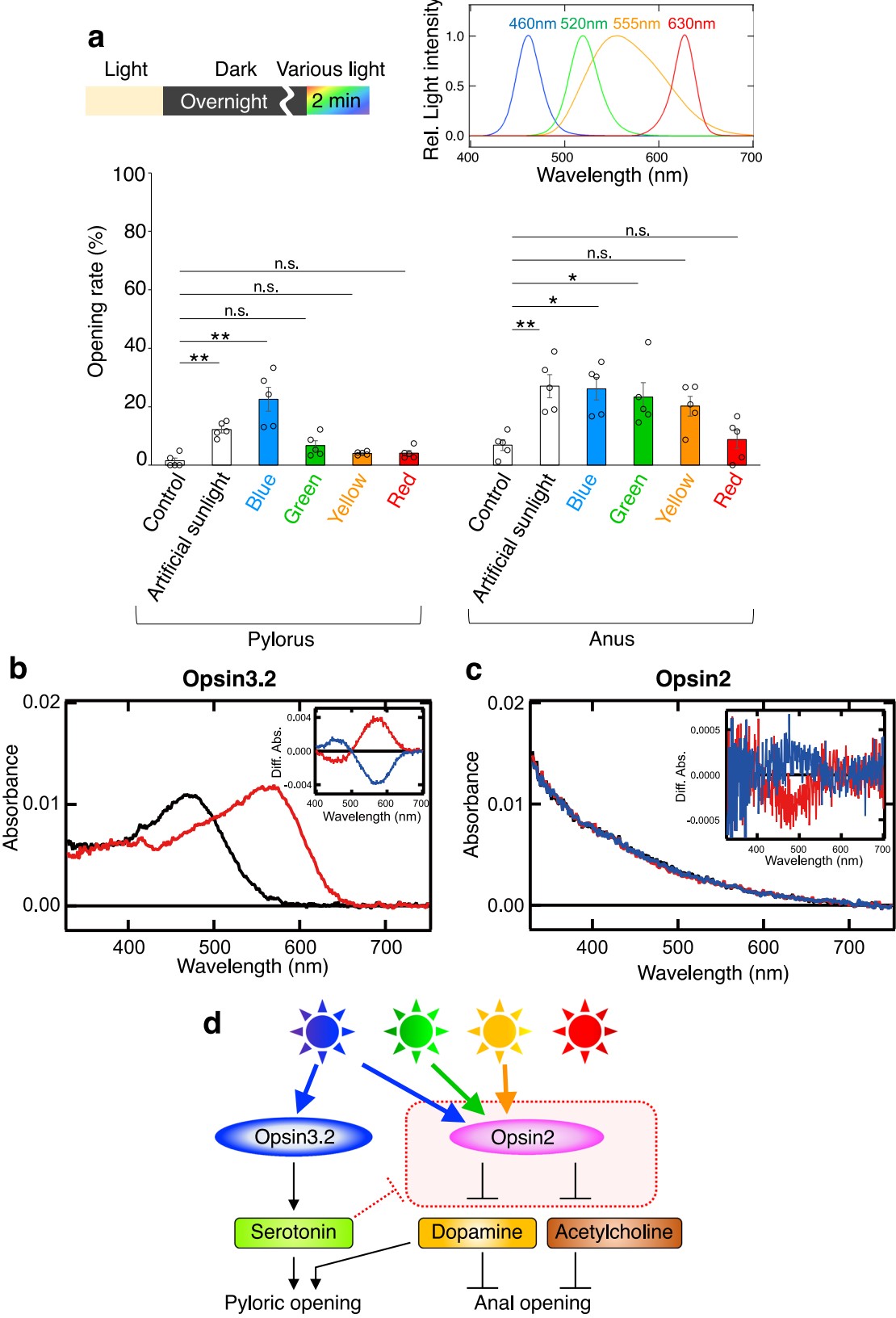

mechanism[17]. The integration of this mechanism with gastrointestinal activity is particularly fascinating.

The evolution of a through-gut in bilaterians represents a significant innovation, offering a sophisticated digestive system with enhanced nutrient absorption efficiency e.g. ref. [2]. This development stands in stark contrast to cnidarians, a sister clade to bilaterians,

which utilize a single orifice for both ingestion and expulsion (Fig. 7)[1,43,44]. The cnidarian gastrovascular cavity, serving dual roles in digestion and nutrient absorption, is inherently less efficient due to its singular cavity design[44]. Additionally, comparisons with the digestive tissues of Xenacoelomorphs, purportedly a clade within bilaterians, yield similar observations. The digestive tissues/organs of

**Fig. 6 | Wavelength dependency in the regulation of pyloric and anal openings.**
**a** Differential regulation of pyloric and anal openings by blue and longer wavelengths, respectively. The inset in the upper right corner details the wavelengths of LED lights utilized in the experiments, and the photon flux density of all LED lights and artificial sunlight was adjusted to 500 μmol m$^{-2}$ s$^{-1}$ ($N$ = 5 batches [each consisting of a different male and female pair], $n$ [control] = 37, 81, 39, 41, 13 larvae, $n$ [artificial sunlight] = 34, 33, 18, 43, 21 larvae, $n$ [blue] = 23, 45, 30, 51, 33 larvae, $n$ [green] = 30, 26, 23, 41, 19 larvae, $n$ [yellow] = 28, 60, 23, 41, 24 larvae, $n$ [red] = 37, 26, 24, 27, 39 larvae; pylorus: [control] mean 1.5% ± 1% SEM, [artificial sunlight] mean 12.2% ± 1.1% SEM, [blue] mean 22.6% ± 4.1% SEM, [green] mean 6.7% ± 1.7% SEM, [yellow] mean 4.1% ± 0.3% SEM, [red] mean 4.1% ± 0.9% SEM, anus: [control] mean 6.9% ± 1.8% SEM, [artificial sunlight] mean 27% ± 4% SEM, [blue] mean 26.2% ± 3.8% SEM, [green] mean 23.3% ± 4.9% SEM, [yellow] mean 20.2% ± 3.3% SEM, [red] mean 8.7% ± 3.2% SEM). **b** Absorption spectra of purified Opsin3.2 measured before (black curve) or after (red curve) blue light (460 nm) irradiation. (inset) Spectral changes

of Opsin3.2 caused by blue light (460 nm) irradiation (red curve) and subsequent red light (>600 nm) irradiation (blue curve). These mirror-image changes were obtained using DDM-solubilized cell membranes expressing Opsin3.2 after the addition of 11-*cis* retinal. **c** Absorption spectra of purified Opsin2 measured before irradiation (black curve) or after yellow light (>500 nm) (red curve) and subsequent violet light (420 nm) (blue curve) irradiation. (inset) Spectral changes of purified Opsin2 caused by yellow light irradiation (red curve) and subsequent violet light irradiation (blue curve). The expression of Opsin2 in cultured cells was also confirmed by Western blot using Rho1D4 antibody (Supplementary Fig. 8). **d** Illustration of the distinct pathways activated by blue light leading to pyloric opening, and by longer wavelengths leading to anal opening, demonstrating wavelength-specific regulatory mechanisms. We used one-way ANOVA followed by Tukey's post hoc test in **a**. Statistical significance denoted as *$p < 0.05$, **$p < 0.01$; n.s. = not significant. Error bars shown in all graphs indicate SEM. Source data are provided as a Source Data file.

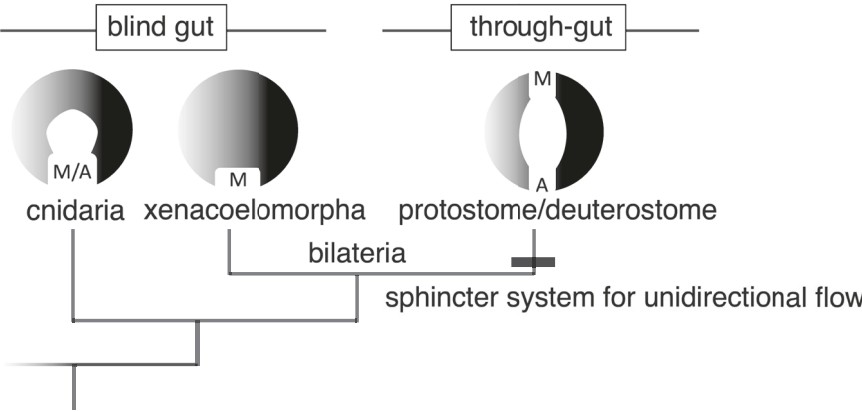

**Fig. 7 | Evolution of through-gut.** The acquisition of a through-gut in the stem lineage of bilaterians was accompanied by the evolution of a sphincter regulatory system, facilitating the unidirectional flow of digestive contents. This development contrasts with cnidarians, which utilize a single orifice for both ingestion and

expulsion, and Xenacoelomorphs, characterized by a blind gut. Additionally, it is intriguing to consider that this sphincter regulatory mechanism might have been influenced by external environmental stimuli, such as light.

Xenacoelomorphs is characterized primarily by an amphistomic blind gut, rather than a definitive through-gut[8,45,46]. As such, it is conceivable that their regulatory mechanisms for the digestive system fundamentally differ from those of protostomes and deuterostomes (Fig. 7).

Regarding the through-gut present in protostomes and deuterostomes, while theories such as the amphistomy scenario and the planuloid-acoeloid scenario suggest diverse evolutionary origins from a common ancestor[47,48], the mechanisms found in the simple through-gut of sea urchin larvae provide crucial insights into features potentially acquired by ancestors of deuterostomes, at least. Continuing this comparative approach, the digestive tract of hemichordate larvae, also compartmentalized by sphincter muscles into a tripartite gut similar to that observed in sea urchin larvae[49]. The presence of a photoreceptor system and similar neural cell groups near the apical organ in hemichordates strongly suggests that Ambulacrarians possess a conserved light-digestive tract interaction function, providing critical support for the notion of functional integration across these systems[49,50].

Moreover, as sphincter coordination is presumed to be an essential function of the through-gut, it is highly likely that this trait was acquired concurrently with the evolution of a fully penetrative digestive tract. In bilaterians, the through-gut facilitates unidirectional flow from ingestion to defecation, allowing for specialized regions for digestion and absorption, such as the stomach and intestine[2]. The emergence of a functional through-gut in bilaterians necessitated the concurrent evolution of a regulatory system for sphincter control to prevent uncoordinated expulsion of ingested material[4,31,51–53]. This likely involved the evolution of a complex nervous system for unilateral sphincter regulation. Our data suggest that this regulatory mechanism may influence the opposing sphincter's closure, indicating

a sophisticated interplay between the two. This mutual regulation likely contributed to the evolution of bilaterians' efficient nutrient absorption systems and complex body structures[31,52,54].

The discovery that sea urchin larvae regulate digestive functions with light underscores evolutionary and practical implications. Given the strong link between circadian rhythms and digestive activity in vertebrates[55–58], where intense light can reset these rhythms, this adaptation suggests new methods for controlling defecation and appetite across species through light exposure. Such insights could lead to novel treatments for digestive disorders and enhanced livestock feeding strategies, highlighting the significant role of environmental cues in digestive system evolution and offering valuable applications in medicine and agriculture.

## Methods
### Animal collection and embryo/larva culture
Adults of *Hemicentrotus pulcherrimus* were collected around Shimoda Marine Research Center, University of Tsukuba, around the Marine and Coastal Research Center, Ochanomizu University, and Research Center for Marine Biology, Tohoku University. Adult sea urchins were collected under the special harvest permission of prefectures and Japan Fishery cooperatives. Gametes were collected by the *intrablastocoelic* injection of 0.5 M KCl, and the embryos/larvae of *H. pulcherrimus* were cultured at 15 °C in glass beakers or plastic dishes that contained filtered natural seawater (FSW) with 50 μg/ml of kanamycin. To observe larval defecation, we fed SunCulture algae (*Chaetoceros calcitrans*, Marinetech, Aichi, Japan, approx. 30,000 cells/μl). An artificial sunlight (XC-500A [370–780 nm], SERIC Ltd., Saitama, Japan) was used for photoirradiation experiments, and the photon flux density

was measured (Apogee Instruments, Logan, UT, USA, or SEKONIC spectromaster C-7000, Tokyo, Japan) and adjusted to 1000 μmol m$^{-2}$ s$^{-1}$ (1/2 ~ 1/3 sunlight of the surface of the sea). For blue, green, yellow, and red light, corresponding LEDs were used for each wavelength and the photon flux density was adjusted to 500 μmol m$^{-2}$ s$^{-1}$ (Fig. 6a). To create dark conditions, the dishes containing larvae were simply wrapped with aluminum foil and kept in dark incubators until use.

## Chemical treatments

The inhibitors and neurotransmitters were diluted in FSW to 10 times final concentration and 100 μl each reagent was applied to 900 μl culture SW 10 sec before photoirradiation. All the following reagents were applied to inhibit specific pathways, and the data support their effectiveness, although we cannot completely rule out the possibility of non-specific interference with other neuronal pathways. Future biochemical experiments on sea urchins will help clarify the specificity of each reagent. Ketanserin tartrate (#153680, FUJIFILM Wako pure Chemical Co., Osaka, Japan) were applied to culture (final 10 μM) to inhibit the serotonin/monoamine pathway. Although Ketanserin tartrate has the potential to non-specifically inhibit other monoamine pathways besides serotonin, a previous study indicated that the effects of this inhibitor were consistent with those observed in serotonin receptor morphants[14]. This suggests that Ketanserin tartrate is likely to specifically inhibit the serotonin pathway in sea urchins. Atropine sulfate (#011-23731, FUJIFILM), previously reported to inhibit muscarinic acetylcholine receptors in sea urchins[17], was applied to culture (final 10 or 100 μM) to inhibit the acetylcholine pathway. Amisulpride (#A2729, Sigma-Aldrich, St. Louis, MO, USA) was applied to culture (final 10, 50 or 100 μM) to inhibit the dopamine pathway, and it has been reported as an effective inhibitor of the dopamine pathway in sea urchins in a previous study[20]. Eticlopride (#E101, Sigma-Aldrich) was applied to FSW (final 10 μM) as an inhibitor for the dopamine pathway[59] (Supplementary Fig. 6). Its effects were comparable to those of Amisulpride (Supplementary Fig. 5), indicating that it is a highly potent inhibitor of dopamine receptors in sea urchins. Serotonin (#S0370, Tokyo Chemical Industry, Tokyo Japan) was dissolved to distilled-water (10 mM) just before use and applied to culture (final 10 μM). Acetylcholine chloride (#00509-31, Nacalai tesque INC., Kyoto, Japan) was dissolved to distilled-water (10 mM) just before use and applied to culture (final 1 or 10 μM). Dopamine (#H8502, Sigma-Aldrich) was dissolved to distilled-water (10 mM) just before use and applied to culture (final 0.1 or 1.0 μM).

## Immunohistochemistry

Whole-mount immunohistochemistry was performed as described previously[60] with some modifications. The 3.7% formaline-fixed samples were blocked with 1% skim milk in PBST for 1 hour at RT and incubated with primary antibodies (dilutions: mouse anti-Synaptotagmin B (SynB)[30] 1:100, rabbit anti-Troponin-I (TnI)[11] 1:200, mouse anti-Opsin2[17] 1:200, rabbit anti-ChAT[17] 1:200, and mouse anti-TH (#22941, Immunostar 1:200) overnight at 4 °C.

## Microinjection of morpholino anti-sense oligonucleotides (MO)

For microinjection, we used injection buffer (24% glycerol, 20 mM HEPES pH 8.0 and 120 mM KCl). The morpholino (Gene Tools, Philomath, OR, USA) sequences and the in-needle concentration with injection buffer were as follows:

Opsin2-MO1 (0.4 mM): 5'- AGTTTGCCATCTTTGTGTTGCTTCG −3' (the specificity was previously confirmed[17]),

Opsin2-MO2 (0.4 mM): 5'- CGCCAATAACCACTGATCACAGTCG −3' (the specificity was previously confirmed[17]),

Opsin3.2 MO1 (0.8-1.0 mM): 5'- ATCTTCTTGAATATGCTTCCG CGCC −3' (the specificity was previously confirmed[14]),

Gcm-MO (1.0 mM), 5'-GCTTTGGACTAACCTTTTGCACCAT −3' (the specificity was previously confirmed[14]),

ChAT-MO (0.5 mM): 5'- ACGATTAGGCATGTGGTTCATGTAT −3' (the specificity was previously confirmed[17]), and

TH-MO (0.5 mM): 5'- TCTTCGGGTTATCTTCCGCCATCGA −3' (the specificity was confirmed by immunostaining (Supplementary Fig. 4b)).

Microinjections into fertilized eggs were performed as previously described[61]. After microinjection, the eggs were washed with FSW three times and stored with 50 μg/ml of kanamycin until the desired stages.

## Microscopy and image analysis

The specimens were observed using a fluorescence microscope (IX73, Evident, Tokyo, Japan) and a confocal laser scanning microscope (FV10i, Evident). All transmission images were taken with the IX73 and the digital camera (DP74, Evident). Panels and drawings for the figures were made using Adobe Photoshop and Microsoft PowerPoint.

## Statistical analysis

No statistical methods were used to predetermine the sample sizes. All n numbers are described in the figure legends. To compare the two groups in Fig. 1b ([anal opening 0 min vs 2 min] $t$-value = −4.5042, dgrees of freedom [d.f.] = 3.3685, $p$-value = 0.0159, [pyloric opening vs anal opening][0 min] $t$-value = 2.3947, d.f. = 5.5061, $p$-value = 0.05741, [1 min] $t$-value = 1.6673, d.f. = 4.044, $p$-value = 0.17, [2 min] $t$-value = 3.4175, d.f. = 3.108, $p$-value = 0.03972, [3 min] $t$-value = 6.1844, d.f. = 3.8882, $p$-value = 0.003813, [4 min] $t$-value = 5.34, d.f. = 4.0635, $p$-value = 0.005667, [5 min] $t$-value = 4.1918, d.f. = 3.4764, $p$-value = 0.01843, [6 min] $t$-value = 3.0045, d.f. = 5.0011, $p$-value = 0.02994, [7 min] $t$-value = 3.6174, d.f. = 3.015, $p$-value = 0.03602, [8 min] $t$-value = 2.4115, d.f. = 3.0187, $p$-value = 0.09436, [9 min] $t$-value = 2.877, d.f. = 3.0797, $p$-value = 0.06172, [10 min] $t$-value = 5.8673, d.f. = 3.116, $p$-value = 0.008874), Fig. 1c ([pylorus] $t$-value = 2.9537, d.f. = 5.9647, $p$-value = 0.02567, [anus] $t$-value = −2.9218, d.f. = 5.5565, $p$-value = 0.02907), Fig. 1d, ([pylorus] $t$-value = −9.8175, d.f. 3.385, $p$-value = 0.001328, [anus] $t$-value = −0.84315, d.f. = 5.5005, $p$-value = 0.4343), Fig. 1e ([pylorus] $t$-value = −5.5664, d.f. = 2.4267, $p$-value = 0.01963, [anus] $t$-value = 5.6323, d.f. = 2.3598, $p$-value = 0.02045), Fig. 2b ([pylorus] $t$-value = −1.232, d.f. = 5.7064, $p$-value = 0.2663, [anus] $t$-value = 2.731, d.f. = 5.9459, $p$-value = 0.03446), Fig. 2c ([pylorus] $t$-value = 0.70794, d.f. = 3.4424, $p$-value = 0.5239, [anus] $t$-value = −16.184, d.f. = 2.8069, $p$-value = 0.0007414), Fig. 2d ([pylorus] $t$-value = −1.5023, d.f. = 11.956, $p$-value = 0.159, [anus] $t$-value = 8.0021, d.f. = 8.2952, $p$-value = 3.547e-05), Fig. 5a ([pylorus] $t$-value = −0.8744, d.f. = 5.8266, $p$-value = 0.4165, [anus] $t$-value = 6.8791, d.f. = 3.0804, $p$-value = 0.005776), Fig. 5b ([pylorus] $t$-value = 1.3417, d.f. = 7.4329, $p$-value = 0.2192, [anus] $t$-value = −4.1479, d.f. = 5.4063, $p$-value = 0.007549), Fig. 5c ([pylorus][Atropine-, Amisulpride-, serotonin+ vs Atropine +, Amisulpride-, serotonin +] $t$-value = 1.3229, d.f. = 2.0153, $p$-value = 0.316, [Atropine-, Amisulpride-, serotonin+ vs Atropine-, Amisulpride +, serotonin +] $t$-value = 39.347, d.f. = 2.732, $p$-value = 7.636e-05, [Atropine-, Amisulpride +, serotonin- vs Atropine-, Amisulpride +, serotonin +] $t$-value = −0.51314, d.f. = 3.5622, $p$-value = 0.638, [anus] [Atropine-, Amisulpride-, serotonin- vs Atropine-, Amisulpride-, serotonin +] $t$-value = 0.28187, d.f. = 3.5557, $p$-value = 0.7937, [Atropine +, Amisulpride-, serotonin- vs Atropine +, Amisulpride-, serotonin +] $t$-value = 0.98652, d.f. = 3.04, $p$-value = 0.3957, [Atropine-, Amisulpride +, serotonin- vs Atropine-, Amisulpride +, serotonin +] $t$-value = −1.3285, d.f. = 3.1478, $p$-value = 0.2721), Supplementary Fig. 1a ($t$-value = −8.0905, d.f. = 15.776, $p$-value = 3.4324e-05), Supplementary Fig. 2a ([pylorus] $t$-value = 4.276, d.f. = 7.5615, $p$-value = 0.003076, [anus] $t$-value = −0.60215, d.f. = 6.8314, $p$-value = 0.5665), Supplementary Fig. 2b ([pylorus] $t$-value = 3.1796, d.f. = 4.9869, $p$-value = 0.02464, [anus] $t$-value = −2.2615, d.f. = 5.4171, $p$-value = 0.06914), Supplementary Fig. 2c ([pylorus] $t$-value = 3.0337, d.f. = 7.6967, $p$-value = 0.01697, [anus] $t$-value = 1.1016, d.f. = 6.8664, $p$-value = 0.3077), Supplementary Fig. 3a ([pylorus] $t$-value = 0.47633, d.f. = 3.1671, $p$-value = 0.6648, [anus] $t$-value = −0.78816, d.f. = 3.5396, $p$-value = 0.48), Supplementary Fig. 3b ([pylorus] $t$-value = −0.11796, d.f. = 3.1514, $p$-

value = 0.9132, [anus] $t$-value = 4.7037, d.f. = 2.1698, $p$-value = 0.0361), Supplementary Fig. 3c ([pylorus] $t$-value = −0.37477, d.f. = 2.6223, $p$-value = 0.7361, [anus] $t$-value = 12.181, d.f. = 2.4253, $p$-value = 0.00311), Supplementary Fig. 6a ([pylorus] $t$-value = 8.446, d.f. = 2.2257, $p$-value = 0.009855, [anus] [Eticlopride- serotonin- vs Eticlopride- serotonin + ]$t$-value = −0.42168, d.f. = 2.0459, $p$-value = 0.7134, [Eticlopride+ serotonin- vs Eticlopride+ serotonin + ]$t$-value = 0.22745, d.f. = 2.7033, $p$-value = 0.8362), we used Welch's $t$-test (two-tailed) with a significance level of 0.001, 0.01 or 0.05. To compare more than two groups, we used one-way ANOVA followed by Tukey's post hoc test with a significance level of 0.01 or 0.05, and the following $F$ values (F) and d.f. For Fig. 6a: pylorus, F = 16.0693, d.f. = 5, [control vs artificial sunlight] $p$-value = 0.0077905, [control vs blue] $p$-value = 0.0010053, [control vs green] $p$-value = 0.433616, [control vs yellow] $p$-value = 0.8999947, [control vs red] $p$-value = 0.8999947; anus, F = 5.9054, d.f. = 5, [control vs artificial sunlight] $p$-value = 0.0072632, [control vs blue] $p$-value = 0.0109105, [control vs green] $p$-value = 0.0383802, [control vs yellow] $p$-value = 0.1366207, [control vs red] $p$-value = 0.8999947, Supplementary Fig. 3d: pylorus, F = 0.3435, d.f. = 2, [atropine 0 μM vs 10 μM] $p$-value = 0.7236509, [atropine 0 μM vs 100 μM] $p$-value = 0.7836383; anus, F = 0.4191, d.f. = 2, [atropine 0 μM vs 10 μM] $p$-value = 0.1021416, [atropine 0 μM vs 100 μM] $p$-value = 0.0084058, Supplementary Fig. 3e: pylorus, F = 5.2509, d.f. = 2, [atropine 0 μM vs 10 μM] $p$-value = 0.8999947, [atropine 0 μM vs 100 μM] $p$-value = 0.0730698; anus, F = 2.3948, d.f. = 2, [atropine 0 μM vs 10 μM] $p$-value = 0.3156187, [atropine 0 μM vs 100 μM] $p$-value = 0.1711409, Supplementary Fig. 3g: pylorus, F = 3.8596, d.f. = 2, [acetylcholine 0 μM vs 1 μM] $p$-value = 0.8999947, [acetylcholine 0 μM vs 10 μM] $p$-value = 0.1235414; anus, F = 10.781, d.f. = 2, [acetylcholine 0 μM vs 1 μM] $p$-value = 0.8441761, [acetylcholine 0 μM vs 10 μM] $p$-value = 0.8999947, Supplementary Fig. 3h: pylorus, F = 3.1631, d.f. = 2, [acetylcholine 0 μM vs 1 μM] $p$-value = 0.1639424, [acetylcholine 0 μM vs 10 μM] $p$-value = 0.1049653; anus, F = 90.7746, d.f. = 2, [acetylcholine 0 μM vs 1 μM] $p$-value = 0.0010053, [acetylcholine 0 μM vs 10 μM] $p$-value = 0.0010053, Supplementary Fig. 5a: pylorus, F = 3.0935, d.f. = 2, [dopamine 0 μM vs 0.1 μM] $p$-value = 0.4606884, [dopamine 0 μM vs 1 μM] $p$-value = 0.0800466; anus, F = 17.4951, d.f. = 2, [dopamine 0 μM vs 0.1 μM] $p$-value = 0.0018365, [dopamine 0 μM vs 1 μM] $p$-value = 0.0014294, Supplementary Fig. 5b: pylorus, F = 2.6978, d.f. = 2, [dopamine 0 μM vs 0.1 μM] $p$-value = 0.709079, [dopamine 0 μM vs 1 μM] $p$-value = 0.1090449; anus, F = 26.2860, d.f. = 2, [dopamine 0 μM vs 0.1 μM] $p$-value = 0.0010053, [dopamine 0 μM vs 1 μM] $p$-value = 0.0010053, Supplementary Fig. 5c: pylorus, F = 0.7099, d.f. = 3, [amisulpride 0 μM vs 10 μM] $p$-value = 0.8999947, [amisulpride 0 μM vs 50 μM] $p$-value = 0.6509181, [amisulpride 0 μM vs 100 μM] $p$-value = 0.8999947; anus, F = 5.3930, d.f. = 3, [amisulpride 0 μM vs 10 μM] $p$-value = 0.339452, [amisulpride 0 μM vs 50 μM] $p$-value = 0.0198519, [amisulpride 0 μM vs 100 μM] $p$-value = 0.013382, Supplementary Fig. 5d: pylorus, F = 4.4781, d.f. = 3, [amisulpride 0 μM vs 10 μM] $p$-value = 0.2869254, [amisulpride 0 μM vs 50 μM] $p$-value = 0.0306875, [amisulpride 0 μM vs 100 μM] $p$-value = 0.0417768; anus, F = 3.9752, d.f. = 3, [amisulpride 0 μM vs 10 μM] $p$-value = 0.8999947, [amisulpride 0 μM vs 50 μM] $p$-value = 0.0341195, [amisulpride 0 μM vs 100 μM] $p$-value = 0.8301446.

### Analysis of the molecular properties of opsin protein

The full-length ORF of Opsin2 cDNA was tagged with the epitope sequence of the anti-bovine rhodopsin monoclonal antibody Rho1D4 (ETSQVAPA) at the C-terminus and was introduced into the mammalian expression vector pCAGGS[62]. The full-length ORF of Opn3.2 cDNA was tagged with the epitope sequence of Rho1D4 at the C-terminus after truncation of C-terminal 157 amino acid residues to improve the expression level[63] and was introduced into pCAGGS. To obtain the recombinant proteins of opsins, the plasmid DNA was transfected into HEK293S cells using the calcium phosphate method. 11-*cis* retinal was added to the medium (final concentration: 5 μM) 24 h after transfection and the cells were kept in

the dark before they were collected at 48 h after transfection. The reconstituted pigments were extracted from cell membranes with 1% dodecyl maltoside (DDM) in Buffer A (50 mM HEPES, 140 mM NaCl, pH 6.5) and were purified using Rho1D4-conjugated agarose. The purified pigments were eluted with 0.02 % DDM in Buffer A containing the synthetic peptide with the epitope sequence. All of the procedures were carried out on ice under dim red light. UV/Vis absorption spectra were recorded with a UV-visible spectrophotometer (UV-2600 or UV-2450, Shimadzu). Samples were kept at 0 °C using a cell holder equipped with a temperature-controlled circulating water bath. The samples were irradiated with light through a KL-42 interference filter, a BP-460 band-pass filter, a Y-52 cutoff filter or an R-60 cutoff filter from a 1 kW tungsten halogen lamp (Master HILUX-HR; Rikagaku).

The activation of Gi-type of G protein by Opsin3.2 was measured by GDP/GTPγS exchange of G protein using a radionucleotide filter-binding assay[64]. Giαβγ was prepared by mixing rat Giα1 expressed in *E. coli* strain BL21[65] with Gtβγ purified from bovine retina[66]. All of the assay procedures were carried out at 0 °C. The assay mixture consisted of 600 nM G protein, 50 mM HEPES (pH 7.0), 140 mM NaCl, 5 mM MgCl₂, 1 mM DTT, 0.01% DDM, 1 μM [$^{35}$S]GTPγS and 2 μM GDP in addition to the purified 20 nM Opsin3.2. Purified Opsin3.2 was mixed with G protein solution and was kept in the dark or irradiated for 1 min with blue light through a BP-460 band-pass filter or with subsequent orange light through an R-60 cutoff filter. After irradiation, the GDP/GTPγS exchange reaction was initiated by the addition of [$^{35}$S]GTPγS solution to the mixture of Opsin3.2 and G protein. After incubation for the selected time in the dark, an aliquot (20 μL) was transferred from the sample into 200 μL of stop solution (20 mM Tris/Cl (pH 7.4), 100 mM NaCl, 25 mM MgCl2, 1 μM GTPγS and 2 μM GDP), and it was immediately filtered through a nitrocellulose membrane to trap [$^{35}$S]GTPγS bound to G proteins. The amount of bound [$^{35}$S]GTPγS was quantitated by assaying the membrane with a liquid scintillation counter (Tri-Carb 2910 TR; PerkinElmer).

The elevation of Ca$^{2+}$ level by Opsin2 in HEK293S cells was measured using an aequorin-based bioluminescence assay[67]. HEK293S cells were seeded in 96-well plates at a density of 60,000 cells/well in medium (D-MEM/F12 containing 10 % FBS). After incubation for 24 h, the plasmid DNA was transfected into the cells (100 ng/well) using the polyethyleneimine transfection method. After incubation for 6 h, 11-*cis* retinal was added to the medium (final concentration: 5 μM). After overnight incubation, the medium was replaced with the equilibration medium (CO₂-independent medium containing coelenterazine h and 10 % FBS). Following equilibration for 2 h at room temperature, luminescence from the cells was measured using a microplate reader (SpectraMax L, Molecular Devices). The cells were irradiated for 5 sec with blue light through a BP-470 band-pass filter or green light through a KL-53 interference filter to trigger the change of the luminescence. To confirm that the HEK293S used to detect the absorption spectrum of Opsin2 truly expresses sea urchin Opsin2 conjugated with a Flag-tag, we performed a Western blot using an anti-Flag antibody. The detailed method for the Western blot was described in the legend of Supplementary Fig. 8.

### Reporting summary

Further information on research design is available in the Nature Portfolio Reporting Summary linked to this article.

## Data availability

Sequence data can be found in the genome database of *Hemicentrotus pulcherrimus*, HpBase (http://cell-innovation.nig.ac.jp/Hpul/)[68]. Source data are provided with this paper.

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

## Acknowledgements

We thank Y. Nakajima, R.D. Burke, and H. Tanaka for essential reagents. We thank M. Kiyomoto, D. Shibata, M. Ooue, J. Takano, Y. Uchida, G. Northen, H. Abe, M. Washio, M. Yamaguchi, and JF Izu/Shimoda for collecting and keeping the adult sea urchins. We thank Prof. Robert S. Molday for the generous gift of a Rho1D4-producing hybridoma, Prof. Jeremy Nathans for providing the HEK293S cell line and Shion Aoki for technical support in the Ca²⁺ assay. This work is supported by JST PRESTO Grant number JPMJPR194C, JST A-STEP Grant number JPMJTR204E, JSPS KAKENHI Grant number 23K23933, the Toray Science Foundation and Takeda Science Foundation to S. Yaguchi. AMED CREST Grant number 22gm1510007 and Research Foundation for Opto-Science and Technology to T. Yamashita.

## Author contributions

Studies were designed by J.Y. and S.Y. Preliminary data were acquired by J.Y. and S.Y. Sea urchin data were acquired by J.Y., T.Yamamoto. and S.Y. Culture cell data were acquired by K.S., A.H., and T.Yamashita. The manuscript was written by J.Y., T.Yamamoto., T.Yamashita. and S.Y. with input from all authors.

## Competing interests

The authors declare no competing interests.
