## [Peer Review file · Nature Communications]

Light-modulated neural control of sphincter regulation in the evolution of through-gut

Corresponding Author: Professor Shunsuke Yaguchi

Version 0:

Reviewer comments:

Reviewer #1

(Remarks to the Author)

The authors address very important questions about the origin, evolution, and regulation of through-gut in bilaterians and metazoans in general. They chose sea urchin larvae as a reference model in this quest. Also importantly, they look at the relationship between light and neurotransmitter regulation (serotonin, ACh, and dopamine) in the control and coordination of gut dynamics.

The presented pharmacological and labeling data about light-transmitter control of sphincter regulation are innovative and noteworthy results. However, I have an impression of an incomplete and rather superficial story in terms of the identification of specific signaling pathways and even neuronal organization in putative circuits controlling these processes. Overall, the paper does not resolve any story or hypotheses related to the early metazoan origins of the gut, and it is better to focus the narrative on deuterostomes or echinoderms with more details. My specific comments and recommendations are summarized below.

Abstract:

The second sentence is too general, and references to ctenophores (#6,7) are out of the context. I suggest a better focus story on deuterostomes with the sea urchin as a reference species.

Main:

The first 1.5 pages are too general without sufficient specifics. Again, here references can be expanded to more specific and original studies. For example, I do not understand ref #12 in the current context. It is rather a speculative review than an original study.

I suggest referring to original studies about basal bilaterian phylogeny (and prebilaterian animals).

I also suggest referring to specific and extensive literature about details of serotonergic and dopaminergic cells in echinoderms and deuterostomes in general as the foundation for the current study. This includes studies of photoreception in adults and larvae, as well as relationships between larval and adult bodyplans in echinoderms.

Experimental chapters:

My major concern is how specific serotonergic, dopaminergic, and cholinergic pathways IN ECHINODERMS AND SEA URCHIN, the respective inhibitors, are used by the authors. Specifics here are crucial since many canonical inhibitors in vertebrates do not work for various invertebrate groups. Moreover, some vertebrate inhibitors have different affinity and specificity to other pathways.

Second, it is important to present actual circuits in greater mechanistic details, controlling both photoreception and muscle control. The authors do not report the specific number of putative serotonergic, dopaminergic, and cholinergic neurons, and photoreceptors. Their morphology and potential interactions are unclear. The small figures with images do not provide sufficient information. I recommend making more figures and specifically providing high-resolution imaging of cells with the respective transmitter systems to show their structures and, ideally, neuronal processes and their overlaps, as well as relationships with muscles.

I also suggest making a more detailed schematic diagram with a more natural representation of photoreceptive and effector cells.

The authors use the term brain for these larvae. Specific structural and functional details are needed (and refs) to justify this term, considering the diffuse character of numerous neurons in the larvae.

The echinoderm larvae have ciliated control of their locomotion. Thus, the authors' speculations/hypotheses about the ecological significance of feeding modes and control of the digestive tract need more specific details with potential incorporation of the ciliated effectors in their schematic diagrams.

Evolution of a through-gut:

The too-general speculations about basal metazoans and mammals in this section are not sufficient and are not based on the factual information presented here. My suggestion is to focus on echinoderms and deuterostomes (including hemichordates) with specific references to basal bilaterians (*Xenoturbella* and acoels) as an outgroup. The current Fig 3e is not informative.

Here, a more specific discussion about the interactions between photoreception and motor control would be useful.

Reviewer #2

(Remarks to the Author)

This well-written paper reports experiments demonstrating that the pyloric and anal sphincter muscles governing the opening of the intestine entry and exit in sea urchin larvae are regulated by light-induced neural pathways. They show that the regulation occurs when light arrives, presumably in the morning in nature, and also show that different wavelengths of light regulate two opsins that trigger the neural activity. The story is well documented with inhibitors of several neural cell types to show that serotonin is the primary regulator of the pyloric opening, a finding that was demonstrated in an earlier paper. Here they add that the anal opening is inhibited by Ach and dopamine input, and that activation of opsin 2 inhibits the Ach and dopaminergic neurons to allow opening of the sphincter. Further, they show that the two sphincters are not open simultaneously but instead are open at separate times to control the passage of detritus in a one-way manner.

They cast the story in an evolutionary context to indicate that this kind of regulation is necessary in all organisms that have a through gut, and indicate, at least in this case, that the one-way movement is regulated by light.

My one question is probably the simplest of all. They provide a wealth of statistical support for every experiment in their study. The biological assay reports the number of occurrences of a closure phenomenon observed. They describe that as a rate and in percentages. Until I got to the supplementary information it wasn't clear how many embryos were observed to produce a percentage. I'm guessing that their "rate" is percent per unit time. Over what time? And in thinking about it, nowhere is it described how the numbers were accumulated. Did they sit at a microscope, turn on the light and rapidly record X number of observed openings per Y min? Did they do it one larva at a time? That isn't described and is key to the entire study. To me it is important to give the reader a feel for how the essential data was recorded.

Beyond that I had two moments of confusion that can easily be corrected editorially:

Pg 4 Line 12. For clarity indicate that the response is to light. You say within 10 minutes, but within 10 min of what? In reading it several times I conclude it is to light, but it would help to make it easier for the reader.

Pg 4 Line 19. To me a better segue would help. When I started this sentence I didn't know whether you were referring to past work or to work reported in this manuscript. Often, in *Nature*, the device used is to say: "Here, we report....."

Reviewer #3

(Remarks to the Author)

Yaguchi et al., Light-modulated neural control of sphincter regulation in the evolution of through-gut: insights from larval sea urchins.

This manuscript focuses on the observation that sea urchin larvae initiate defecation in response to light, and builds upon previous work that demonstrated that light can induce pyloric sphincter opening. The authors utilize an immunohistochemical approach to quantify the opening rates (percentages of samples) that show pyloric or anal opening and utilize pharmacology and knock-down experiments to elucidate components of the pathways involved in mediating the light-dependent opening of both the pyloric and anal sphincters. Their findings will be of interest to those researching

echinoderm neurobiology (an under-explored field) but also more broadly to researchers of animal enteric nervous system regulation, the evolution of gut regulation across phyla, and circadian rhythm-mediated modulation of behavior. The experimental approaches they use are appropriate and the work is strengthened by their integration of behavioral, pharmacological, and biochemical data to support their conclusions.

In the manuscript, the authors contrast the light-modulation of anal opening with that of pyloric modulation, and the opposing actions of the serotonergic system on these two events in which serotonin increases pyloric opening but decreases anal opening. They identify Opsin2 as an opsin involved in light-mediated anal opening, further contrasting anal opening with pyloric opening that is mediated by Opsin3.2, and they all perform some basic biochemical characterization of these opsins to link their absorption profiles with sphincter behavior and the hypothesized evolutionary benefits of these two opsins being utilized. Using exogenously applied neurotransmitters and inhibitors they determine that anal opening is inhibited by the dopaminergic and cholinergic systems, and that this inhibition is removed by activation of Opsin2 by broad spectrum light. They confirmed from their previous study (Yaguchi, J., & Yaguchi, S. (2021). *BMC biology*, 19(1), 64. <https://doi.org/10.1186/s12915-021-00999-1>) that the pyloric opening is mediated by the serotonergic system, and they further determined that there is cross talk between the two pathways mediating pyloric and anal opening.

Although the majority of the experiments are logical and lead to straightforward conclusions, I have two small concerns with the experimental results. 1) Although 50 μ M amisulpride is shown to significantly increase anal opening in Fig. 2f, no such increased effect of 50 μ M amisulpride is shown in extended data Fig. 4d. This may be because of the low n numbers in these experiments (2-3) precluding statistical analysis or the initial high rate of anal opening in the absence of amisulpride in Fig. 4d, and I recommend increasing the size of the data set for these experiments to confirm the effects shown in Fig. 2. The principle applies to Fig. 2e and extended Fig. 4b with regards to dopamine. 2) The absorbance data for Opsin2 (Fig. 3c) is striking in that there is no peak within the visible light spectrum and the spectrum changes very little upon irradiation. How can the authors be sure that their samples genuinely contained functional Opsin2?

I also have several minor comments that should be straightforward to respond to and address.

Minor comments

1. I'm not sure you need "on Earth" in the opening sentence of the main text.
2. Page 5, line 8. Please clarify which specific data for anal and pyloric opening was significantly different, e.g. entire time period, just in light, first three time points.
3. Page 5, line 24. Please add a reference for the expression of Opsin 2 in sea urchin larval arms.
4. Page 6, line 18. Missing word? I believe this should read "rarely do so simultaneously" or something that similarly expresses the same idea.
5. Fig. 1 legend annotation should read "I" for intestine not "I" to match Fig. 1a.
6. For extended data Fig. 1a, please clarify methodological details pertaining to the sizes and numbers of fields of view used in the quantification of excrement counts.
7. Extended data Fig. 1a. Use a consistent number of decimal places for the y axis numbering.
8. Where any statistical tests done on the data on extended Fig. 4? If so, these tests and their results should be reported.
9. Ketanserin, which is used here as an inhibitor of serotonin receptors, is also an inhibitor of the relatively non-specific vesicular neurotransmitter transporter VMAT2 (Erickson, J. D., Schafer, M. K., Bonner, T. I., Eiden, L. E., & Weihe, E. (1996). *Proceedings of the National Academy of Sciences of the United States of America*, 93(10), 5166–5171. <https://doi.org/10.1073/pnas.93.10.5166>). Although this does not significantly affect their results and conclusions, this should be mentioned in the text as a caveat to their experiments involving ketanserin.

Version 1:

Reviewer comments:

Reviewer #1

(Remarks to the Author)

In this revised manuscript, the authors addressed all my major concerns and improved both the logic and details of their study.

In summary, this is an interesting report and an important research direction in comparative and integrative neurobiology.

Minor comments:

Lines 212-214 " within the anal sphincter region, there are one or two anal neurons
These neurons exhibit cholinergic and dopaminergic properties, suggesting a role in the constant suppression of anal opening"

- it is unclear what exactly do you mean "exhibit" cholinergic and dopaminergic "properties"
- What properties? What transmitters are in these neurons? Are dopamine and ACh co-localized?

Reviewer #2

(Remarks to the Author)

This revised manuscript managed to satisfy my concerns in the initial submission. It lays out a sequence of light activated events that leads to opening or closing of two sphincters in the gut of sea urchin larvae. The experiments are clear and quantified well. My only remaining suggestions are several – to help the reader with interpretation of the figures. The figures are fine even though they are quite busy, and with minor additions they can help the reader.

The table was not included in this version but need not be sent separately because it was fine when seen in the original version of the paper and I don't think it was changed in the revision.

1, line 114. The white arrowheads are on Fig. 1b. are seen first. I then realized that the images of the closed and open anal sphincter you talk about are below the cartoon of the larva. To me this figure, therefore, is very busy. It might help if the subsections were separated into boxes. That way, I would immediately realize the white arrowheads are actually in the fig. 1a box. Fig. 1b would also be helped as the graph could be confused otherwise.

2. Line 128. You say you score the opening rate. Is that really the case? Aren't you actually measuring the percent of larvae with open vs closed anal sphincters? To me the rate would be a measure of the dynamics of the movement from closed to open state, and you are actually looking at before vs after and not the actual dynamic.

Fig. 1C. It would help to color your bar graphs either yellow or green to indicate your measurement of anal or pyloric opening. Also, you say, while on this figure, that serotonin inhibits the anal opening – and Fig. 1C suggests this, while Fig. 1D strongly reinforces it experimentally. I suggest including Fig. 1C in that evidence also.

Line 140. I suggest you say “selective” attenuation of opsin 3.2 or something to indicate that your experiment was removal of the opsin rather than both opsin and serotonin.

The table file is missing from the material supplied to this reviewer.

Reviewer #3

(Remarks to the Author)

The authors have addressed all of my initial comments in this revised version of the paper.

Response to Reviews

Reviewer #1 (Remarks to the Author):

The authors address very important questions about the origin, evolution, and regulation of through-gut in bilaterians and metazoans in general. They chose sea urchin larvae as a reference model in this quest. Also importantly, they look at the relationship between light and neurotransmitter regulation (serotonin, ACh, and dopamine) in the control and coordination of gut dynamics.

The presented pharmacological and labeling data about light-/transmitter control of sphincter regulation are innovative and noteworthy results. However, I have an impression of an incomplete and rather superficial story in terms of the identification of specific signaling pathways and even neuronal organization in putative circuits controlling these processes. Overall, the paper does not resolve any story or hypotheses related to the early metazoan origins of the gut, and it is better to focus the narrative on deuterostomes or echinoderms with more details. My specific comments and recommendations are summarized below.

Indeed, we fully agree that the conclusions drawn from this discovery regarding the metazoan gut system are too wide and out of focus. However, the mechanisms we discovered, such as the inability to open both ends simultaneously and the unidirectional transport of gut contents, are crucial for the formation of the through-gut. Because through-gut is a defining feature of bilaterians and is essential for their characterization, the focus should be shifted from discussions on metazoans to bilaterians, or at the very least, deuterostomes.

Abstract:

The second sentence is too general, and references to ctenophores (#6, 7) are out of the context. I suggest a better focus story on deuterostomes with the sea urchin as a reference species.

Thank you very much for your insightful comments. We fully agree with your suggestion to exclude references to ctenophores, as they indeed fall outside the scope of this study. We also believe that highlighting the acquisition of the through-gut by the common ancestor of bilaterians provides essential evolutionary context and emphasizes the broader significance of our findings. In line with the format change, we have extensively modified the abstract to address your concerns while retaining this critical aspect. We will focus the main discussion on bilaterians, using the sea urchin as the reference species, while briefly mentioning the evolutionary milestone of the through-gut in the bilaterian ancestor to set the stage for our study.

We hope this approach meets your approval and look forward to any further suggestions you might have.

Main:

The first 1.5 pages are too general without sufficient specifics. Again, here references can be expanded to more specific and original studies. For example, I do not understand ref #12 in the current context. It is rather a speculative review than an original study.

I suggest referring to original studies about basal bilaterian phylogeny (and prebilaterian animals).

I also suggest referring to specific and extensive literature about details of serotonergic and dopaminergic cells in echinoderms and deuterostomes in general as the foundation for the current study. This includes studies of photoreception in adults and larvae, as well as relationships between larval and adult bodyplans in echinoderms.

Thank you for these suggestions. We agree with the reviewer's comments. We have substantially revised the Introduction (the initial part of the Main section) to address the reviewer's concerns while preserving the key points of our original submission. To avoid misunderstanding and ensure logical coherence, we have not included references to adult sea urchins due to the significant differences between larvae and adults. Additionally, we will discuss photoreception in Results part not in Introduction as it becomes relevant.

Experimental chapters:

My major concern is how specific serotonergic, dopaminergic, and cholinergic pathways IN ECHINODERMS AND SEA URCHIN, the respective inhibitors, are used by the authors. Specifics here are crucial since many canonical inhibitors in vertebrates do not work for various invertebrate groups. Moreover, some vertebrate inhibitors have different affinity and specificity to other pathways.

Thank you for your comment. We agree that the specificity of inhibitors in sea urchins is an important point. We have added a comment acknowledging that we cannot completely rule out the possibility that the reagents may inhibit other neural pathways in the last sentence of the inhibitor experiment section.

Here, we detail the four reagents used in our experiments:

Ketanserin: Used for blocking the serotonin pathway, Ketanserin may non-specifically block other monoamine pathways (as mentioned by reviewer #3).

However, it has been previously reported that Ketanserin produces the same results as serotonin receptor morphants in sea urchins. We have cited the paper in the Methods section. Therefore, we believe it is an effective inhibitor for blocking serotonin signaling in sea urchins.

Atropine: Used for blocking the cholinergic pathway, Atropine is a well-known inhibitor of muscarinic ACh receptors, which are abundantly expressed in the ciliary band of sea urchin larvae. Our previous study demonstrated that Atropine interferes with ciliary beating, confirming its effectiveness in blocking the cholinergic pathway in sea urchin larvae. This study is referenced in the Methods section.

Amisulpride: Used for blocking the dopamine pathway, Amisulpride has been previously reported to inhibit the dopamine pathway in sea urchins in a study from the other laboratory. This reference is included in the Methods section.

Eticlopride: Used for blocking the dopamine pathway, Eticlopride has been previously reported to inhibit the dopamine pathway in sea urchins in a study from the other laboratory, and its effects were comparable to those of Amisulpride by our hands (*cf.* Supplementary Fig. 5e *vs* Supplementary Fig. 6b), indicating that it is a highly potent inhibitor of dopamine receptors in sea urchins. This information is also referenced in the Methods section.

Based on these points, while we cannot entirely exclude the possibility of non-specific effects of these reagents, we conclude that the inhibitors used are largely specific to each pathway. We have also attempted knockdown experiments for each pathway, but most of these affected multiple developmental and physiological events, making them unsuitable for analyzing timely gut function. Consequently, these inhibitors are the sole choice currently available. These points have been described in the Methods section to maintain the logical flow of the main text.

We modified “Chemical treatments” with proper references in Method section to:

Chemical treatments

The inhibitors and neurotransmitters were diluted in FSW to 10 times final concentration and 100 μ l each reagent was applied to 900 μ l culture SW 10 sec before photoirradiation. All the following reagents were applied to inhibit specific pathways,

and the data support their effectiveness, although we cannot completely rule out the possibility of non-specific interference with other neuronal pathways. Future biochemical experiments on sea urchins will help clarify the specificity of each reagent. Ketanserin tartrate (FUJIFILM Wako pure Chemical Co., Osaka, Japan) were applied to culture (final 10 μM) to inhibit the serotonin/monoamine pathway. Although Ketanserin tartrate has the potential to non-specifically inhibit other monoamine pathways besides serotonin, a previous study indicated that the effects of this inhibitor were consistent with those observed in serotonin receptor morphants¹⁴. This suggests that Ketanserin tartrate is likely to specifically inhibit the serotonin pathway in sea urchins. Atropine sulfate (FUJIFILM), previously reported to inhibit muscarinic acetylcholine receptors in sea urchins¹⁷, was applied to culture (final 10 or 100 μM) to inhibit the acetylcholine pathway. Amisulpride (Sigma-Aldrich, St. Louis, MO, USA) was applied to culture (final 10, 50 or 100 μM) to inhibit the dopamine pathway, and it has been reported as an effective inhibitor of the dopamine pathway in sea urchins in a previous study²⁰. Eticlopride (Sigma-Aldrich) was applied to FSW (final 10 μM) as an inhibitor for the dopamine pathway⁶⁰ (Supplementary Fig. 6). Its effects were comparable to those of Amisulpride (Supplementary Fig. 5), indicating that it is a highly potent inhibitor of dopamine receptors in sea urchins. Serotonin (Tokyo Chemical Industry, Tokyo Japan) was dissolved to distilled-water (10 mM) just before use and applied to culture (final 10 μM). Acetylcholine chloride (Nacalai tesque INC., Kyoto, Japan) was dissolved to distilled-water (10 mM) just before use and applied to culture (final 1 or 10 μM). Dopamine (Sigma-Aldrich) was dissolved to distilled-water (10 mM) just before use and applied to culture (final 0.1 or 1.0 μM).

Second, it is important to present actual circuits in greater mechanistic details, controlling both photoreception and muscle control. The authors do not report the specific number of putative serotonergic, dopaminergic, and cholinergic neurons, and photoreceptors. Their morphology and potential interactions are unclear. The small figures with images do not provide sufficient information. I recommend making more figures and specifically providing high-resolution imaging of cells with the respective transmitter systems to show their structures and, ideally, neuronal processes and their overlaps, as well as relationships with muscles.

I also suggest making a more detailed schematic diagram with a more natural representation of photoreceptive and effector cells.

Thank you for your comment. We strongly agree with your suggestions and have incorporated schematic images to illustrate the details of neural positions and the relationship between neurons and muscular sphincters in Fig. 2d. Additionally, we have added a schematic image in Fig. 2i to depict the signal flow from photoreceptors to neurons, although some details remain unclear.

To present high-resolution neural patterns in the larval endoderm, we have included a new Supplementary Figure 4, which shows cholinergic and dopaminergic neurons identified by in situ hybridization and immunohistochemistry, along with schematic images. We also included ChAT⁻ and TH-morphants to demonstrate the specificity of the immunostaining (as previously mentioned, knockdown experiments produced multiple effects, so we did not use them for the functional analysis of each neuron at the timing we focused on).

We have noted the cell number of neurons in Supplementary Fig. 4 and in main text with corresponding reference as the reviewer suggested.

The authors use the term brain for these larvae. Specific structural and functional details are needed (and refs) to justify this term, considering the diffuse character of numerous neurons in the larvae.

In the new version of the Introduction, we briefly describe why we use "brain" in the current/recent studies, supported by relevant references in Pg 4. "In the anterior part of the larvae, defined as the brain based on its gene expression profile and its role in integrating external stimuli and transducing these signals into behavioral responses^{13,14}, serotonergic and cholinergic neurons are present^{15,16}."

The echinoderm larvae have ciliated control of their locomotion. Thus, the authors' speculations/hypotheses about

the ecological significance of feeding modes and control of the digestive tract need more specific details with potential incorporation of the ciliated effectors in their schematic diagrams.

Thank you for the suggestion. The coordination of ciliary beating and feeding in response to light is indeed a fascinating aspect of larval behavior. Consequently, we have included “ Intriguingly, the photoreceptors involved in the opening of the anus in sea urchin larvae are consistent with Opsin2, which halts ciliary movement¹⁷. Typically, sea urchin larvae reside near the ocean's surface, where they coordinate ciliary beating through cholinergic neurons to perform anti-gravity movements, although the detailed mechanisms remain unclear. Opsin2, upon exposure to light, suppresses cholinergic neuronal activity, causing the larvae to sink. This phenomenon has been suggested as a component of the DVM mechanism¹⁷. The integration of this mechanism with gastrointestinal activity is particularly fascinating.”, in the final sentence of the relevant section, supported by an appropriate reference. However, we opted not to include a schematic diagram in the figure, as although this point is intriguing, it does not constitute the main focus of our study.

Evolution of a through-gut:

The too-general speculations about basal metazoans and mammals in this section are not sufficient and are not based on the factual information presented here. My suggestion is to focus on echinoderms and deuterostomes (including hemichordates) with specific references to basal bilaterians (Xenoturbella and acoels) as an outgroup. The current Fig 3e is not informative.

Here, a more specific discussion about the interactions between photoreception and motor control would be useful.

Thank you for your suggestion. We agree and have revised the discussion to focus exclusively on bilaterian gut evolution, removing references to metazoans, including Ctenophora, as mentioned in the Introduction. Figure 3e has been updated accordingly.

Given that the relationship between light and sphincter muscle in gut is a novel discovery, at least to our knowledge, it was challenging to incorporate a detailed discussion on photoreceptors and motor control. We attempted this but found it did not fit well. However, it is well-established that light regulates gut activity through circadian rhythms in a wide variety of organisms. Therefore, we added references for fish and birds because their relationship between light and gut activity is well-documented. We substituted mammal to vertebrates.

We would like to retain the speculative section, as we hope this phenomenon will be investigated in other organisms, especially humans. We believe such speculation will attract a broad range of readers, including non-scientists, in general journals like Nature Communications. We aim to avoid overstating our findings, so the sentences are concise but necessary for context. We hope you understand this point.

Reviewer #2 (Remarks to the Author)

This well-written paper reports experiments demonstrating that the pyloric and anal sphincter muscles governing the opening of the intestine entry and exit in sea urchin larvae are regulated by light-induced neural pathways. They show that the regulation occurs when light arrives, presumably in the morning in nature, and also show that different wavelengths of light regulate two opsins that trigger the neural activity. The story is well documented with inhibitors of several neural cell types to show that serotonin is the primary regulator of the pyloric opening, a finding that was demonstrated in an earlier paper. Here they add that the anal opening is inhibited by Ach and dopamine input, and that activation of opsin 2 inhibits the Ach and dopaminergic neurons to allow opening of the sphincter. Further, they show that the two sphincters are not open simultaneously but instead are open at separate times to control the passage of detritus in a one-way manner.

They cast the story in an evolutionary context to indicate that this kind of regulation is necessary in all organisms that have a through gut, and indicate, at least in this case, that the one-way movement is regulated by light.

My one question is probably the simplest of all. They provide a wealth of statistical support for every experiment in their study. The biological assay reports the number of occurrences of a closure phenomenon observed. They describe that as a rate and in percentages. Until I got to the supplementary information it wasn't clear how many embryos were observed to produce a percentage. I'm guessing that their "rate" is percent per unit time. Over what time? And in thinking about it, nowhere is it described how the numbers were accumulated. Did they sit at a microscope, turn on the light and rapidly record X number of observed openings per Y min? Did they do it one larva at a time? That isn't described and is key to the entire study. To me it is important to give the reader a feel for how the essential data was recorded.

Thank you for this comment and we are sorry for making you confused. We missed to comment how we set the experimental condition and how we measured the ratio of pyloric/anal openings. We add "Therefore, as previously described¹⁴, we set a constant light/dark cycle (10 min light, 16 h dark, and 2 min with or without photoirradiation [photon flux density, 1000 $\mu\text{mol m}^{-2} \text{s}^{-1}$]). We fixed the larvae 2 min after photoirradiation and checked the opening and closing of the pyloric and anal

sphincters using immunohistochemistry with anti-TroponinI, which specifically detects these sphincters. The ratio of pyloric and anal opening/closing events was quantified in the fixed and immunostained larvae.”, to Pg 6 to clarify it. In addition, we added Method section to mention the details.

Beyond that I had two moments of confusion that can easily be corrected editorially:

Pg 4 Line 12. For clarity indicate that the response is to light. You say within 10 minutes, but within 10 min of what? In reading it several times I conclude it is to light, but it would help to make it easier for the reader.

The reviewer is correct. We fixed it to “10 minutes after being transferred to light condition from dark” to clarify what we did.

Pg 4 Line 19. To me a better segue would help. When I started this sentence I didn't know whether you were referring to past work or to work reported in this manuscript. Often, In Nature, the device used is to say: “Here, we report.....

Thank you for this suggestion. We added, “Here, we report that the anus of sea urchin larvae starts to open within 1 to 2 minutes after being irradiated with light.”

Reviewer #3 (Remarks to the Author)

Yaguchi et al., Light-modulated neural control of sphincter regulation in the evolution of through-gut: insights from larval sea urchins.

This manuscript focuses on the observation that sea urchin larvae initiate defecation in response to light, and builds upon previous work that demonstrated that light can induce pyloric sphincter opening. The authors utilize an immunohistochemical approach to quantify the opening rates (percentages of samples) that show pyloric or anal opening and utilize pharmacology and knock-down experiments to elucidate components of the pathways involved in mediating the light-dependent opening of both the pyloric and anal sphincters. Their findings will be of interest to those researching echinoderm neurobiology (an under-explored field) but also more broadly to researchers of animal enteric nervous system regulation, the evolution of gut regulation across phyla, and circadian rhythm-mediated modulation of behavior. The experimental approaches they use are appropriate and the work is strengthened by their integration of behavioral, pharmacological, and biochemical data to support

their conclusions.

*In the manuscript, the authors contrast the light-modulation of anal opening with that of pyloric modulation, and the opposing actions of the serotonergic system on these two events in which serotonin increases pyloric opening but decreases anal opening. They identify Opsin2 as an opsin involved in light-mediated anal opening, further contrasting anal opening with pyloric opening that is mediated by Opsin3.2, and they all perform some basic biochemical characterization of these opsins to link their absorption profiles with sphincter behavior and the hypothesized evolutionary benefits of these two opsins being utilized. Using exogenously applied neurotransmitters and inhibitors they determine that anal opening is inhibited by the dopaminergic and cholinergic systems, and that this inhibition is removed by activation of Opsin2 by broad spectrum light. They confirmed from their previous study (Yaguchi, J., & Yaguchi, S. (2021). *BMC biology*, 19(1), 64.*

<https://doi.org/10.1186/s12915-021-00999-1> that the pyloric opening is mediated by the serotonergic system, and they further determined that there is cross talk between the two pathways mediating pyloric and anal opening.

Although the majority of the experiments are logical and lead to straightforward conclusions, I have two small concerns with the experimental results.

1) Although 50 μM amisulpride is shown to significantly increase anal opening in Fig. 2f, no such increased effect of 50 μM amisulpride is shown in extended data Fig. 4d. This may be because of the low n numbers in these experiments (2-3) precluding statistical analysis or the initial high rate of anal opening in the absence of amisulpride in Fig. 4d, and I recommend increasing the size of the data set for these experiments to confirm the effects shown in Fig. 2. The principle applies to Fig. 2e and extended Fig. 4b with regards to dopamine.

Thank you for your comment. We have increased the sample size and performed statistical analysis on all data presented in Supplementary Figure 5 (previously Extended Figure 4). Our results now clearly demonstrate that Amisulpride at 50 μM and Dopamine at 0.1 μM significantly regulate anal opening. (Regarding the reviewer's comment comparing Figure 2f and Supplementary Figure 5d (originally Extended Figure 4), we believe the intended comparison is with Supplementary Figure 5c (originally Extended Figure 4). There is a significant difference observed with 50 μM Amisulpride to control. We have also increased the sample size and performed statistical analysis for these comparisons as per the reviewer's suggestion. In addition, we added the statistical analysis in the Supplementary Figure 3.

2) The absorbance data for Opsin2 (Fig. 3c) is striking in that there is no peak within the visible light spectrum and the spectrum changes very little upon irradiation. How can the authors be sure that their samples genuinely contained functional Opsin2?

Thank you for this comment. Although the spectral change was very small, the cultured cells expressing Opsin2 exhibited an increase of calcium level by light irradiation. We also confirmed the expression of Opsin2 protein in cultured cells by Western blot analysis. We added the data in Supplementary Fig. 8. We also put the explanation for the experiment in the legend of the figure.

I also have several minor comments that should be straightforward to respond to and address.

Minor comments

1. *I'm not sure you need "on Earth" in the opening sentence of the main text.*

We agree with the reviewer. We deleted it.

2. *Page 5, line 8. Please clarify which specific data for anal and pyloric opening was significantly different, e.g. entire time period, just in light, first three time points.*

Thank you for the indication. We have clarified it by including statistical analysis for each time point between the anal and pyloric openings. We have modified the sentence accordingly to, "The timing of anal opening varied among individuals, and the frequency of anal opening was significantly higher compared to that of the pylorus during 2 to 7 min of the 10-minute observation period (Fig. 1b).".

3. *Page 5, line 24. Please add a reference for the expression of Opsin 2 in sea urchin larval arms.*

We refer our previous paper (Yaguchi et al 2022 PLoS Genet).

4. *Page 6, line 18. Missing word? I believe this should read "rarely do so simultaneously" or something that similarly expresses the same idea.*

Thank you for this. We fixed it to "but they rarely do simultaneously".

5. *Fig. 1 legend annotation should read "I" for intestine not "T" to match Fig. 1a.*

Thank you for this. We fixed it.

6. *For extended data Fig. 1a, please clarify methodological details pertaining to the sizes and numbers of fields of view used in the quantification of excrement counts.*

Thank you for this comment and the reviewer is correct. We missed to describe the methodology here. We added "The larvae were kept in the dark in a ϕ 3.5 cm dish containing 3.0 ml of seawater. We counted the number of excretions in the dish at the moment the light was turned on and again 10 minutes later. " to Supplementary Fig. 1a legend.

7. *Extended data Fig. 1a. Use a consistent number of decimal places for the y axis numbering.*

Thank you for this. We fixed it.

8. *Where any statistical tests done on the data on extended Fig. 4? If so, these tests and their results should be reported.*

Thank you for this. We performed statistical analyses, presented the results in the graphs, and referred to them in the statistical analysis section.

9. *Ketanserin, which is used here as an inhibitor of serotonin receptors, is also an inhibitor of the relatively non-specific vesicular neurotransmitter transporter VMAT2 (Erickson, J. D., Schafer, M. K., Bonner, T. I., Eiden, L. E., & Weihe, E. (1996). Proceedings of the National Academy of Sciences of the United States of America, 93(10), 5166–5171. <https://doi.org/10.1073/pnas.93.10.5166>). Although this does not significantly affect their results and conclusions, this should be mentioned in the text as a caveat to their experiments involving ketanserin.*

Thank you for this comment. We have shown that Ketanserin function was same as knockdown of serotonin receptor (5-HT₂). Therefore, in figure legend of the corresponding figure we mentioned that this reagent is a potent inhibitor for other neurotransmitters as this reviewer commented, and also mentioned that we have shown the 5-HT₂ knockdown experiment in the previous study with citations. In addition, as the reviewer #1 commented we added the sentence that all reagents cannot completely exclude the possibility of non-specific inhibitory effects on other neural pathways.